# Wide-spread brain activation and reduced CSF flow during avian REM sleep

Gianina Ungurean [1,8] ✉, Mehdi Behroozi [2,8] ✉, Leonard Böger[3,4], Xavier Helluy[2,5], Paul-Antoine Libourel [6], Onur Güntürkün [2,7] & Niels C. Rattenborg[1]

Mammalian sleep has been implicated in maintaining a healthy extracellular environment in the brain. During wakefulness, neuronal activity leads to the accumulation of toxic proteins, which the glymphatic system is thought to clear by flushing cerebral spinal fluid (CSF) through the brain. In mice, this process occurs during non-rapid eye movement (NREM) sleep. In humans, ventricular CSF flow has also been shown to increase during NREM sleep, as visualized using functional magnetic resonance imaging (fMRI). The link between sleep and CSF flow has not been studied in birds before. Using fMRI of naturally sleeping pigeons, we show that REM sleep, a paradoxical state with wake-like brain activity, is accompanied by the activation of brain regions involved in processing visual information, including optic flow during flight. We further demonstrate that ventricular CSF flow increases during NREM sleep, relative to wakefulness, but drops sharply during REM sleep. Consequently, functions linked to brain activation during REM sleep might come at the expense of waste clearance during NREM sleep.

Sleep is a dangerous state of reduced environmental awareness thought to perform important functions for the brain[1]. Recently, mammalian sleep has been implicated in maintaining a healthy extracellular environment in the brain[2]. During wakefulness, neuronal activity leads to the accumulation of toxic proteins, such as amyloid-β and tau, implicated in Alzheimer's disease[3–5]. The recently described glymphatic system[6] is thought to rectify this problem[2,7]. Specifically, cerebrospinal fluid (CSF) flows into the perivascular space along arteries penetrating the brain and enters the extracellular space, facilitated by the water channel aquaporin 4 (AQP4) on astrocytic endfeet[8], where it mixes with the interstitial fluid, and then drains via venous perivascular spaces, removing waste in the process[2]. Interestingly, as shown in mice, this process occurs during non-rapid eye

movement (NREM) sleep[2]. In addition, although CSF flow into the brain has not been measured directly in sleeping humans, increased ventricular CSF flow, visualized using functional magnetic resonance imaging (fMRI), during NREM sleep is thought to be coupled to flow through the glymphatic system[9].

Several aspects of sleep's proposed role in clearing waste from the brain remain unresolved. Notably, it is unclear how CSF flow changes throughout the ventricular system and brain from NREM to REM sleep[7,10–13], a paradoxical state with wake-like brain activity, during which we experience our most vivid, bizarre, story-like, and emotional dreams[14]. It is also unknown whether waste removal mediated by CSF flow through the brain is a general function of sleep shared by mammalian and non-mammalian

[1]Avian Sleep Group, Max Planck Institute for Biological Intelligence, Seewiesen, Germany. [2]Department of Biopsychology, Institute of Cognitive Neuroscience, Faculty of Psychology, Ruhr-University Bochum, Bochum, Germany. [3]Max-Planck Research Group Neural Information Flow, Max Planck Institute for the Neurobiology of Behavior – caesar, Bonn, Germany. [4]Max-Planck Research Group Genetics of Behaviour, Max Planck Institute for the Neurobiology of Behavior – caesar, Bonn, Germany. [5]Department of Neurophysiology, Medical Faculty, Ruhr-University Bochum, Bochum, Germany. [6]CRNL, SLEEP Team, UMR 5292 CNRS/U1028 INSERM, Université Claude Bernard Lyon 1, Lyon, Bron, France. [7]Research Center One Health Ruhr, Research Alliance Ruhr, Ruhr-University Bochum, Bochum, Germany. [8]These authors contributed equally: Gianina Ungurean, Mehdi Behroozi. ✉e-mail: gianina.ungurean@bi.mpg.de; mehdi.behroozi@ruhr-uni-bochum.de

species. Birds are well suited to address these questions. Despite last sharing a common ancestor with mammals over 300 million years ago, birds exhibit sleep states remarkably similar to mammalian NREM and REM sleep[15,16]. Although episodes of avian REM sleep are short (typically <10 s) when compared to mammals, birds, such as pigeons, engage in hundreds of episodes per night. Importantly, birds are also the only non-mammalian group to express AQP4 on astrocytic endfeet facing the blood vessels, a necessary cellular component of the glymphatic system[7,17]. Finally, as birds are homeotherms with high metabolic rates and large brains more densely packed with neurons than in mammals[18,19], they might have a greater need to clear metabolic waste from the brain.

Here we combined fMRI and pupillometry in awake and naturally sleeping pigeons to examine state dependent changes in brain activation and ventricular CSF flow. Although we had initially planned to examine only the former, as we were collecting our data, Fultz and colleagues[9] reported that ventricular CSF flow can be visualized using fMRI. Consequently, our focus shifted to also examine state-specific changes in the CSF signal. We show that brain regions involved in processing the visual world during wakefulness are activated during avian REM sleep. We further demonstrate that, as in mammals, ventricular CSF flow is greater during NREM sleep when compared to wakefulness. And, importantly, we show that REM sleep is coupled to a sharp drop in CSF flow throughout the ventricular system. Consequently, our findings suggest that a trade-off exists between functions performed by wide-spread brain activation during REM sleep and waste clearance during NREM sleep. Alternatively, we propose that despite impeding the flow of CSF in the ventricular system, the influx of blood into the brain during REM sleep might play an unexpected, complimentary role in moving fluids through the brain.

## Results

### Pigeons rapidly fall asleep inside the fMRI scanner and cycle frequently between NREM and REM sleep

Before examining sleep state-dependent changes in CSF flow, we first established that avian REM sleep is associated with widespread brain activation. We measured whole brain BOLD from naturally sleeping pigeons ($n = 15$) in a 7 T fMRI scanner and simultaneously acquired video recordings through an MR-compatible infra-red video camera (Fig. 1a). The birds were head-fixed using a chronically implanted plastic pedestal (Fig. 1a) to ensure a low level of head movements throughout the entire recording period. Only 0.2% of the volumes had frame-wise displacement higher than 0.09 mm (~20% of voxels size, Supplementary Figs. 1 and 2). Prior to the fMRI recording sessions, the birds underwent an extensive training procedure for progressive amounts of time (Fig. 1b) to reduce the stress associated with head fixation and scanner acoustic noise[20]. After completing the habituation phase, the last two sessions were independently analyzed, and then combined using fixed effect modeling (Methods). Using well-established behavioral correlates of EEG-defined NREM and REM sleep[15,21] (Supplementary Fig. 3), we then identified all unambiguous bouts of NREM and REM sleep from the video recordings while blind to the corresponding fMRI signals. NREM was scored when both eyelids were closed and the eyes, pupils, and bill were still, whereas REM sleep was scored when both eyelids were closed, but the eyes, pupils, and/or bill were moving rapidly (see Materials and Methods, Movie S1). The birds had on average 85 bouts of NREM sleep (range 23–143) and 72 bouts of REM sleep (range 33–140; Fig. 1c). The number of NREM and REM sleep bouts were not statistically different between sessions ($p_{NREM\ S1/S2} = 0.17$, $p_{REM\ S1/S2} = 0.65$; Fig. 1c).

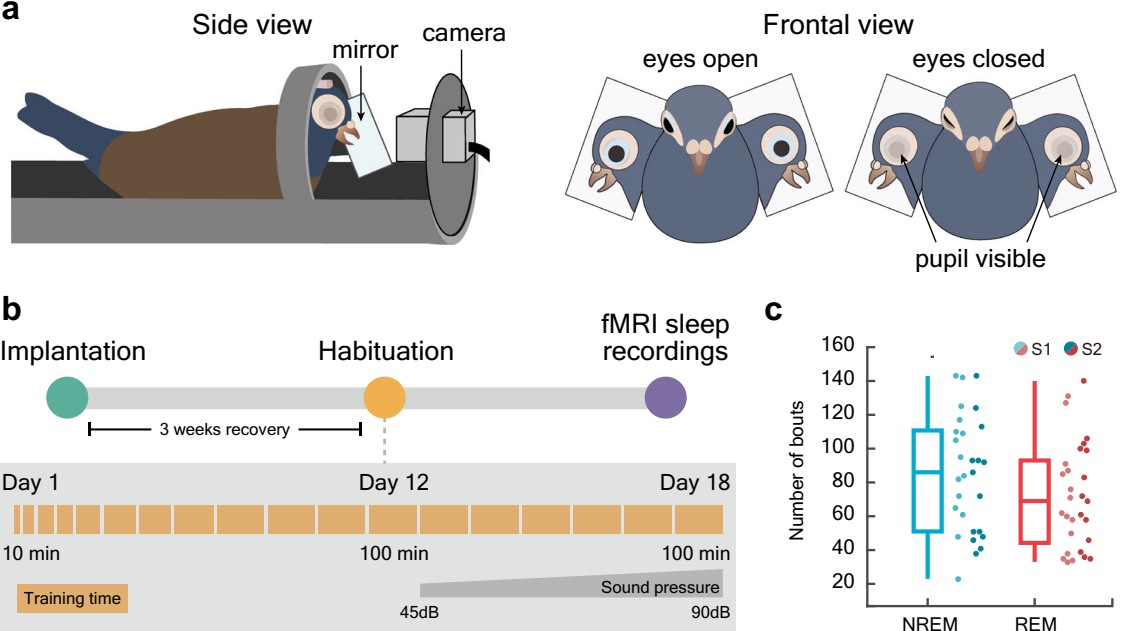

**Fig. 1 | Measuring naturally occurring NREM and REM sleep in pigeons inside a 7 T scanner. a** Experimental setup. Head-restrained pigeons were positioned in the center of the scanner bore and their eyes, pupils, and bill were monitored using an MR-compatible camera. Two mirrors were placed bilaterally to cover both eyes using a single camera. **b** Before running the main experiment, pigeons underwent 18 days of habituation to the scanner setup. **c** To test the reproducibility of the results, two independent sessions were analyzed for each pigeon. Due to low amounts of NREM and REM sleep in the second session of one pigeon, only the first session was included in the analysis (session 1: $n = 15$ pigeons, session 2: $n = 14$ pigeons). The number of NREM and REM sleep bouts were not statistically different between sessions ($p_{NREM\ S1/S2} = 0.17$, $p_{REM\ S1/S2} = 0.65$). The sessions are coded with bright (session 1, S1) and dark (session 2, S2) colors. Each dot represents the number of NREM/REM bouts for a single bird. Boxplots are centered at the median with hinges at 1st and 3rd quartiles and whiskers drawn from hinges to the lowest and highest points within 1 interquartile range. Source data are provided as a Source Data file.

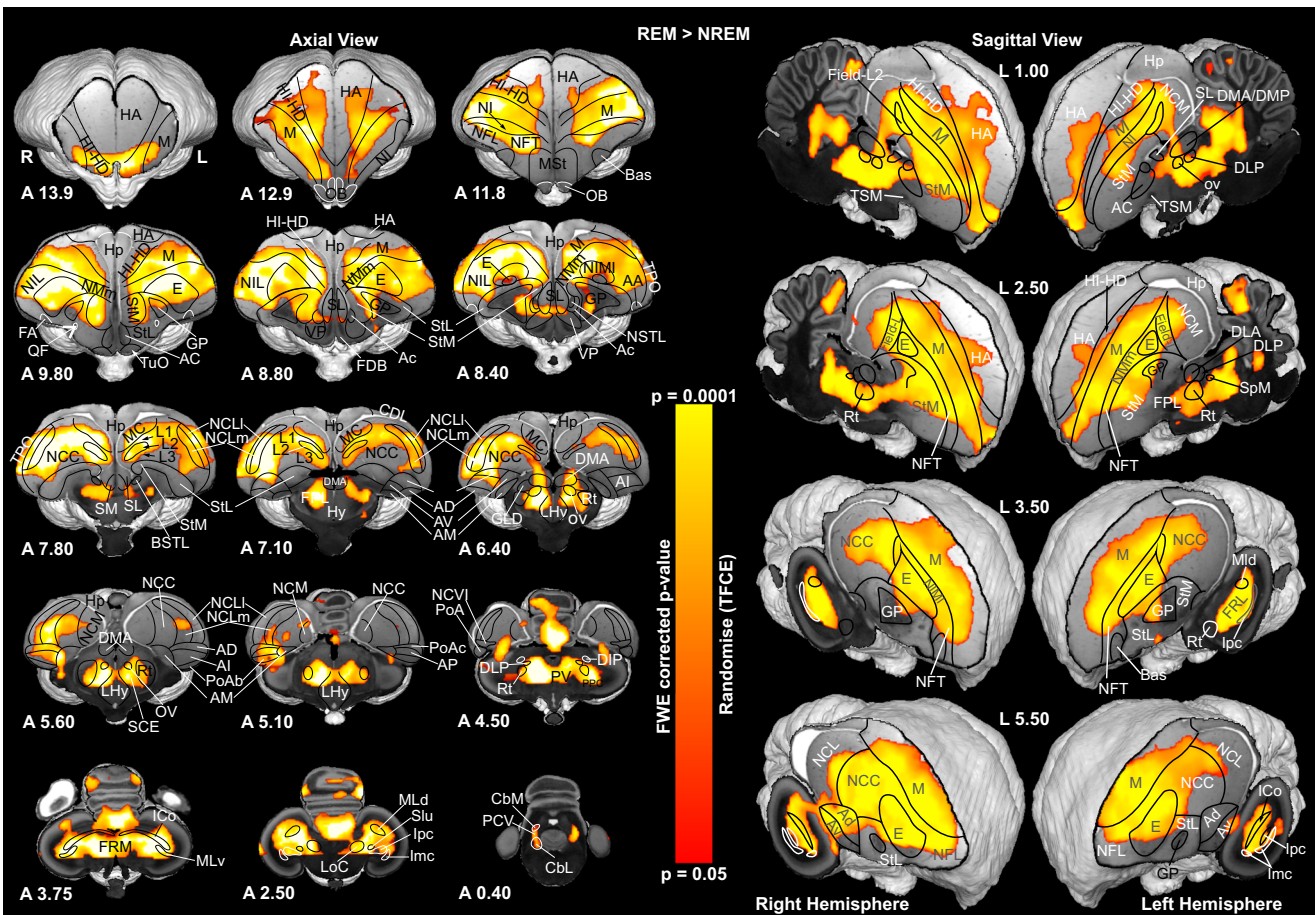

**Fig. 2 | Brain activation patterns during REM sleep.** A GLM analysis was used to demonstrate the activated networks during REM sleep by examining the REM > NREM contrast. The pigeon-specific hemodynamic response function[20] was used to optimize the modeled BOLD responses. The left panel shows the activation pattern of group-averaged data from 15 pigeons (29 sessions) in the axial view (group analysis using a nonparametric permutation-based test, "randomise", 5000 permutations, with Threshold-Free Cluster Enhancement and a family-wise error correction, corrected for multiple comparison across the whole brain and significance at a threshold of $p < 0.05$). Functional maps were overlaid on the high-resolution anatomical data at the different levels of an ex-vivo pigeon brain (in greyscale). The right panel represents the same activation pattern during REM sleep in the sagittal views. The activation significance is demonstrated by the color scale. Anatomical borders (black and white lines) are based on the contrast difference in the ex-vivo Budapest pigeon brain, the pigeon brain atlas[62], and the telencephalic connectome of the pigeon forebrain[81]. The corresponding abbreviations of delineated ROIs are listed in Supplementary Table 1. Frontal and sagittal slice coordinates are defined based on the Budapest pigeon's brain anatomy (Supplementary Fig. 4). *TFCE* Threshold-Free Cluster Enhancement, *FEW* family-wise error.

## BOLD fMRI responses during REM sleep reveal activity in most of the sensory and multi-sensory networks of the pigeon brain

We investigated the whole-brain activity patterns during REM sleep by analyzing REM > NREM sleep contrasts. As illustrated in Fig. 2, robust BOLD activation patterns within the telencephalon were found in most primary and all secondary sensory processing areas, as well as in multimodal integration regions. Within the hyperpallium, we found prominent activation clusters both in thalamopallial input areas (IHA, HI; see Supplementary Table 1 for abbreviations) as well as in associative structures (HD, HA) of the visual and the somatosensory hyperpallium. Within the dorsal ventricular ridge, primary visual (entopallium) and auditory (field L2) areas were active. In addition, associative areas of the trigeminal (NFT), the tectofugal visual (NIMl, NMm, NIL), and the auditory systems (fields L1, L3, NCM) demonstrated significant BOLD signal increases. We also observed significant activity in the convergence zones of both thalamo- and tectofugal visual pathways (NFL, TPO). In the higher-order mesopallial associative areas, the trigeminal (MFV), visual tectofugal (MIVl, MVL), auditory (MC), multimodal (MIVm), and limbic components (MFD) evinced significant activity patterns[22]. In addition, limbic areas like the nucleus accumbens (AC), parts of the septum laterale (SL), the bed nucleus of the stria terminalis (NSTL), and several subnuclei of the amygdala

showed significant BOLD increases. Similarly, most of the anterior part of the prefrontal-like nidopallium caudolaterale (NCL)[23], the limbic-associative nidopallium caudocentrale (NCC), and the (pre)motor arcopallium (AI, AD) were active. While the hippocampal formation (Supplementary Fig. 5) did not evince significant BOLD signals, the area dorsolateralis corticoidea (CDL) that closely interacts with the hippocampal formation was active[24]. At the subpallial level we found significant BOLD signals in lateral and medial striatum (StM, StL), globus pallidus (GP), and ventral pallidum (VP).

Within the diencephalon, the thalamic relay nuclei of the visual tectofugal (n. rotundus, Rt) and auditory pathways (n. ovoidalis, Ov) were activated along with the lateral hypothalamus and the n. dorsomedialis anterior (DMA) that corresponds to the intralaminar nuclei[25] and participates in the regulation of attention and arousal[26]. The brainstem pontomesencephalic reticular structures evinced significant BOLD responses that partly reached the deep layers of the tectum. Within the cerebellar cortex, we observed significant activity patterns in folia II to VI. We also found activation clusters in the cerebellovestibular process (PCV), the ventral part of the medial cerebellar nucleus (CbM), and the medial part of the lateral cerebellar nucleus (CbL)[27]. All of the results were highly repeatable (Supplementary Fig. 6).

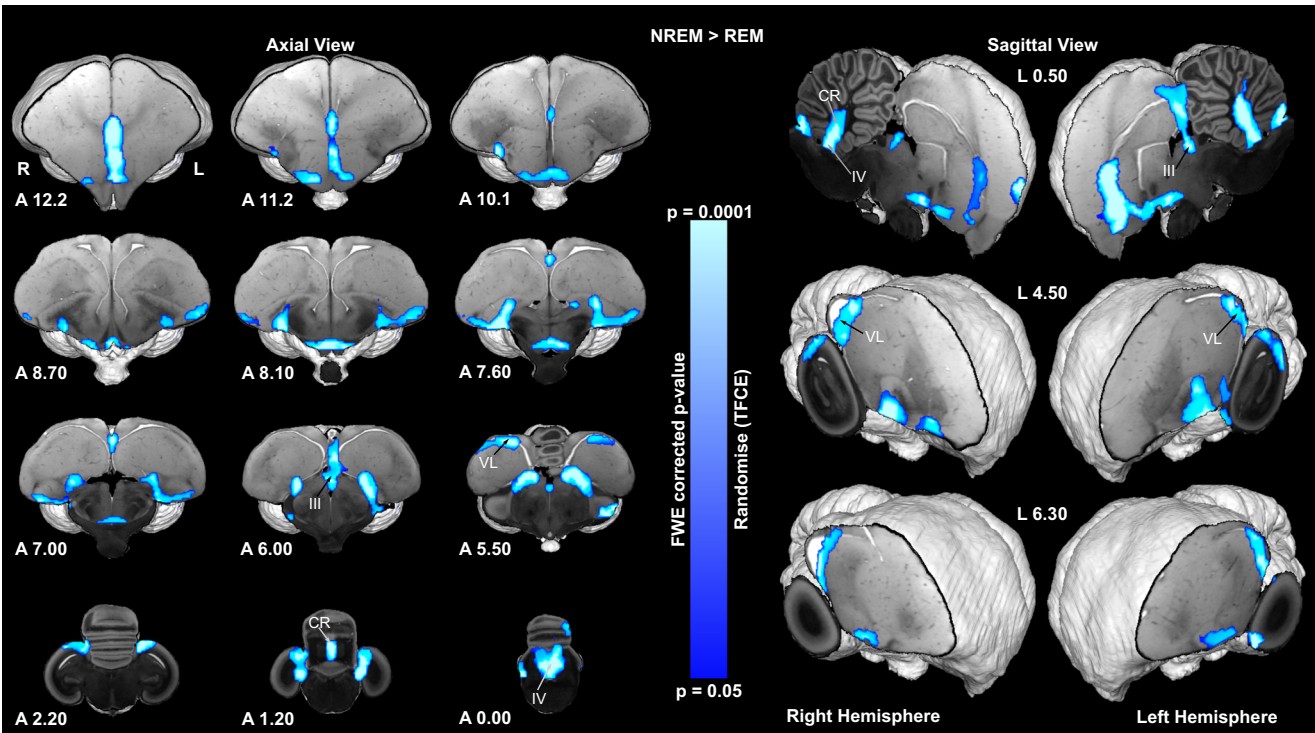

**Fig. 3 | Increased fMRI signal associated with NREM sleep.** The high-resolution axial and sagittal slices at the different levels of an ex-vivo pigeon brain are in greyscale, while the contrast map represents the significant increase of fMRI signal during the NREM sleep (NREM > REM contrast, $n = 15$ pigeons, 29 sessions). Group analysis was done using a nonparametric permutation-based test, "randomise", 5000 permutations, with Threshold-Free Cluster Enhancement and a family-wise error correction. Group statistical maps were corrected for multiple comparisons across the whole brain and significance at a threshold of $p < 0.05$. The significance of the fMRI signal increase is demonstrated by the color scale. The corresponding abbreviations of delineated ROIs are listed in Supplementary Table 1. Frontal and sagittal slice coordinates are defined based on the Budapest pigeon's brain anatomy (Supplementary Fig. 4). TFCE: Threshold-Free Cluster Enhancement; FEW: family-wise error.

## The ventricular system of the pigeon shows increased fMRI signal during NREM sleep

In contrast to REM sleep, most of the significant fMRI signal associated with NREM sleep was located in the ventricular system or in the adjoining margins of the brain (Fig. 3). In the ventricles, it extended along the lateral ventricles englobing the caudal nidopallium, regionally in the third ventricle (III), and in the fourth (IV) ventricle, including its extension into the cerebellum, the cerebellar recess (CR)[28]. In addition, the interhemispheric fissure showed significant fMRI signal (Fig. 3). All results were repeatable across the two sessions (Supplementary Fig. 7). As in previous studies[9,11,29], we interpret these fMRI signals to reflect the inflow of CSF (see Supplementary Material). Finally, a bilateral cluster near the base of the telencephalon extended into the striatum in an area with large penetrating arteries (Fig. 3, A 7.6), reflecting either activation of the neighboring striatum or CSF flow along these arteries.

## Avian REM sleep is associated with BOLD activity surges in the telencephalon (gray matter) and sharp decreases in CSF inflow signal

To further understand the CSF dynamics, we focused on the BOLD-CSF relationship between NREM and REM sleep. The CSF and BOLD time-series were, respectively, extracted from the IV ventricle (excluding the CR) located at the bottom slices (Fig. 4a, green mask), to maximize the sensitivity to the CSF inflow (as in refs. 9 and [29]), and the telencephalon (Fig. 4a, red mask). As shown in a representative time series, the increases in the BOLD signal are linked to changes in the CSF signal, suggesting a strong coupling between the two (Fig. 4b). The cross-correlation results indicated a negative peak with a lag of +4 s (Fig. 4c,

$-0.13$, $p < 0.0001$ permutation test) which resembles a similar shape to that reported in humans[29]. This result confirmed the existence of significant temporal coupling between CSF and BOLD signals. We further calculated the average time course of the CSF and BOLD signals locked to the onset of bouts of NREM and REM sleep, and found that while the CSF flow signal during NREM sleep increased, the gray matter BOLD signal decreased (Fig. 4d). Interestingly, the opposite occurred in REM sleep; as the CSF flow signal decreased, the gray matter BOLD signal increased (Fig. 4e). In addition, the parameter estimates of the IV ventricle and telencephalon were significantly different during NREM and REM sleep (Fig. 4f, g, NREM: $t(14) = -5.3$, $p < 0.0001$, paired-sample t-test; REM: $t(14) = -5.3$, $p < 0.0001$, paired-sample t-test).

## CSF flow decreases during avian wakefulness

Finally, we used two approaches to examine how CSF flow changes during wakefulness (Supplementary Table 2). First, we identified brain regions activated when the birds had one or both eyes open. In birds, including pigeons, unilateral eye closure is associated with an interhemispheric asymmetry in NREM sleep-related EEG slow-wave activity (2–4 Hz power) recorded from electrodes placed over the visual hyperpallium at the dorsal surface of the brain[30–32]. During unilateral eye closure, the hemisphere contralateral to the open eye shows lower slow-wave activity, indicative of a more awake-like state. Birds direct the open eye toward potential threats and react quickly to threatening visual stimuli presented to the open eye, suggesting that asymmetric NREM sleep serves a predator detection function[30,31]. In our fMRI recordings, pigeons with both eyes open showed increased signal in the left and right entopallium, associative areas (NIMl, NMm, NIL), and higher-order mesopallial associative areas (MIVl, MVL) when

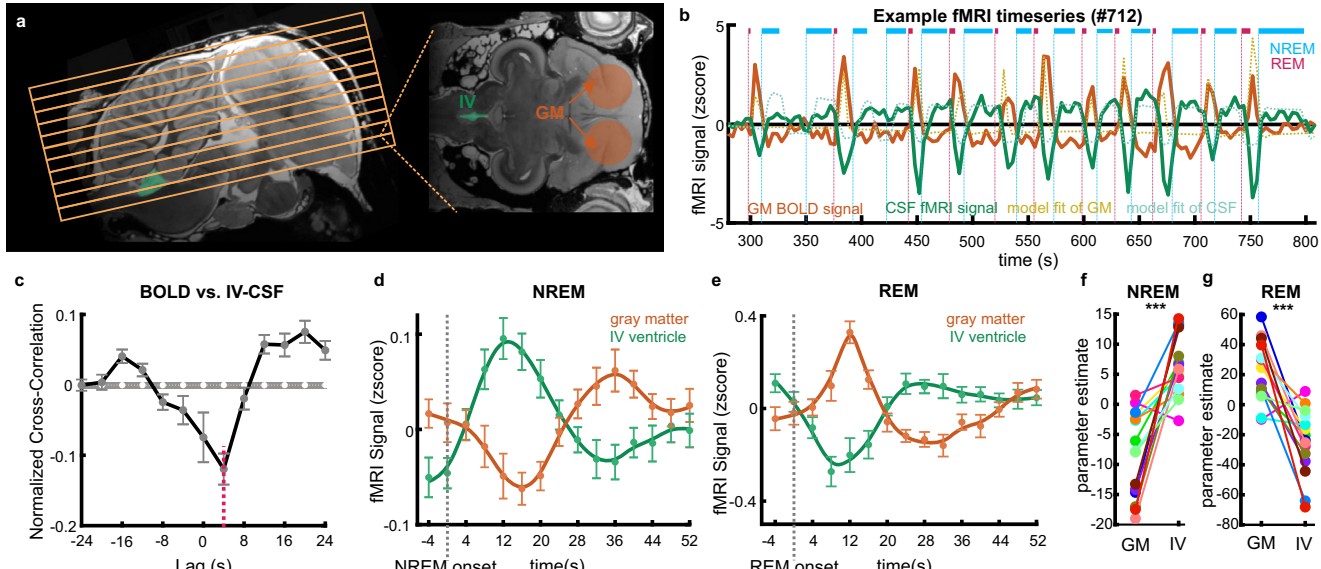

**Fig. 4 | Association between BOLD signal in the telencephalon (gray matter) and CSF inflow signal. a** Position of acquired functional slices (yellow grid) and selected ROIs (orange and green masks) relative to the high-resolution anatomical image. The BOLD signal was extracted from the voxels in the pigeon's telencephalon (orange mask, a sphere with a diameter of 6.5 mm centered at the entopallium to cover most of the telencephalon). The CSF signal was averaged from the IV ventricle (excluding the cerebellar recess), at the bottom slices of the fMRI acquisition (green mask). **b** Representative example time-series of the telencephalic BOLD and CSF signals depicting changes in both signals relative to bouts of NREM and REM sleep. During sleep, the BOLD signal dynamics are anticorrelated to the CSF signal. The BOLD signal increases during REM sleep when CSF flow decreases. The CSF flow increases during NREM sleep when the BOLD signal decreases. **c** Mean cross-correlation between BOLD and CSF signals ($n = 29$, 15 pigeons). The negative cross-correlation at +4 s lag (red dashed line) demonstrates the strongest coupling between BOLD and CSF signal (−0.13, $p = 2e-04$ two-sided permutation test, red dashed line). The white dashed line and gray shaded region represent the mean correlation at each time lag and the 95% confidence interval of shuffled data. Error bars denote the standard error of the mean (SEM) across different lags. **d, e** Average time courses in BOLD and CSF signals locked on NREM and REM sleep onset, respectively ($n = 29$ sessions, 15 pigeons). Error bars represent SEM. **f, g** Parameter estimates during the NREM and REM sleep ($n = 15$). Colored circles show the average beta estimate over the GM mask and IV ventricle mask for each bird. CSF inflow increases compared to the BOLD signal during NREM sleep ($t(14) = 5.4$, $p = 9e-05$, two-tailed paired-sample t-test), and the opposite occurs during REM sleep ($t(14) = −5.3$, $p = 1e-04$, two-tailed paired-sample t-test). Similar results were obtained in the cerebellar recess of the IV ventricle (Supplementary Fig. 8). Source data are provided as a Source Data file.

compared to NREM sleep with both eyes closed (Fig. 5a), suggesting that these visual regions were awake. Opening of only one eye was primarily associated with fMRI signal in the entopallium; when only the left eye was open, the right entopallium was activated (Fig. 5b), whereas when only the right eye was open, activation of the entopallium was bilateral (Fig. 5c). Nonetheless, in both unilateral eye states, activation of the entopallium (Fig. 5d) was significantly greater in the hemisphere contralateral to the open eye (Fig. 5e).

During all states with at least one eye open, the signal in the IV ventricle (Fig. 5f) and CR (Fig. 5g) was significantly lower when compared to NREM sleep, indicating that as in mammals, wakefulness is associated with reduced CSF flow. Nonetheless, these CSF signals were higher when compared to REM sleep. Although this might indicate that CSF flow is lower during REM sleep than wakefulness, the quality of wakefulness in the scanner might underestimate the impact that wakefulness has on CSF flow. As some NREM sleep-related EEG slow-wave activity can occur in a hemisphere when the contralateral eye is open in pigeons[31,33], the brain activation occurring during this eye state in the scanner is likely less than that of fully awake pigeons interacting with their environment.

Our second approach to assess how CSF flow changes during wakefulness, was to present a novel sound stimuli shown to evoke typical waking EEG activity (Supplementary Figs. 4 and 9). As with having one or both eyes open, when compared to NREM sleep, the CSF signal decreased in response to the sound (Fig. S10a, $t(3) = 4.02$, $p < 0.01$, paired sample t-test). A similar BOLD-CSF relationship was found between the gray matter and the cerebellar recess of the IV ventricle (Supplementary Fig. 10b, $p < 0.01$, paired sample t-test).

## Discussion

The widespread BOLD activity pattern during REM-sleep in pigeons shows remarkable similarities to results obtained in humans[34]. We observed significant activations in the mesopontine tegmentum, cerebellum, visual and intralaminar thalamus, amygdalar nuclei, further limbic and paralimbic structures, striatum, parts of the avian prefrontal-like structure, and most sensory and associative sensory areas of the pallium with the majority of these sensory areas being visual. This overlaps with findings in mammals[35–41]. Thus, the bilateral activation of primary and higher order visual regions in pigeons might support visual imagery during REM sleep[38]. Such imagery might be linked to rapid eye movements, which are thought to track visual scenes in dreams[38,42–44]. Activation of the amygdala suggests that such experiences involve emotions[34,43]. The potential emotional content might be linked to the rapid constrictions of the pupils occurring during REM sleep, which also occur during courtship and aggression in awake birds[21]. However, even in humans, it remains controversial whether behaviors occurring during REM sleep are directly linked to dream content[45,46].

It is obviously unclear if pigeons experience dreams similar to humans[47]. With respect to the content of potential dreams, however, a very preliminary hypothesis could be formulated. When organisms move forward, they experience optic flow as a visual expansion in the direction of self-motion[48], while local motion signals code for obstacles that should be avoided in flight through visually cluttered environments[49]. Such optic flow and local motion signals are integrated in the oculomotor cerebellum of which folia VI is a part[50]. Folia II-VI were also activated during REM sleep and further integrate tactile input from the whole-body surface of the bird, including the wings[51].

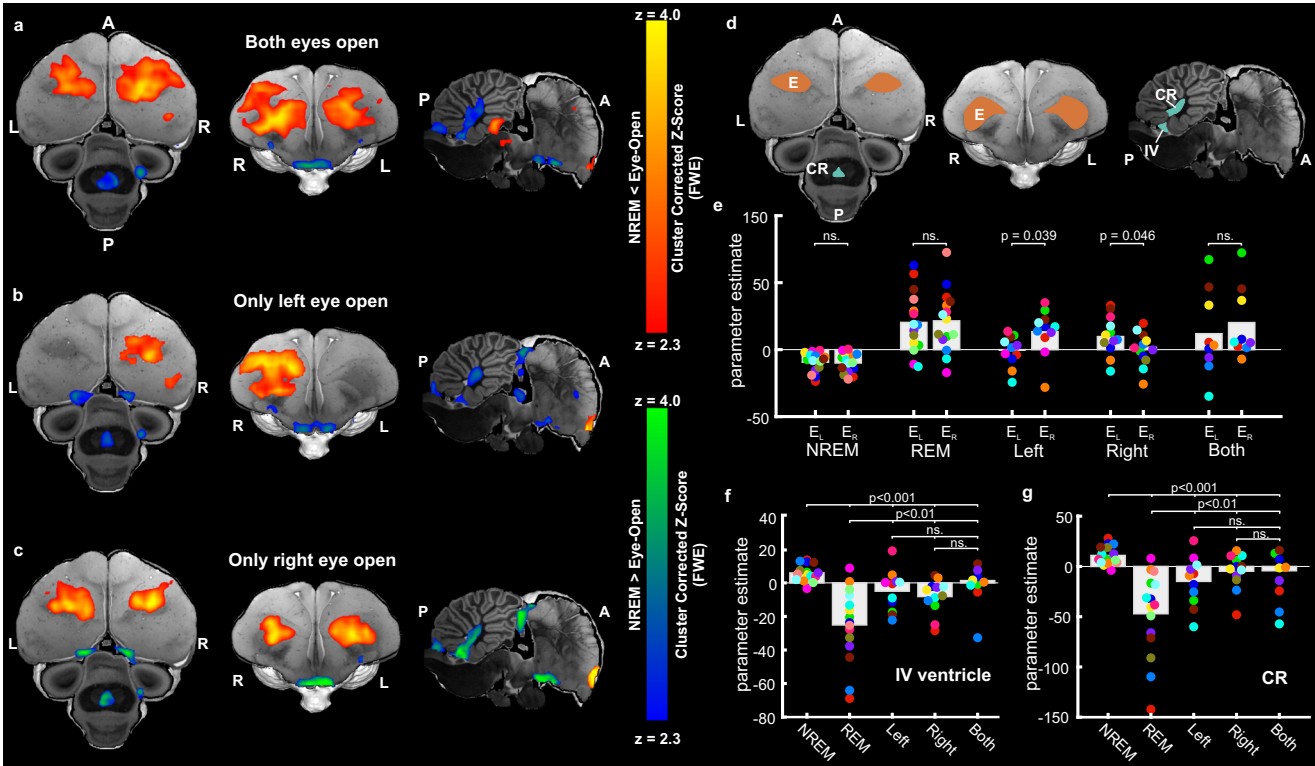

**Fig. 5 | Association between eye-open states and BOLD signal in the telencephalon (gray matter) and ventricular CSF inflow.** GLM analysis was used to demonstrate the activated networks during eye opening by contrasting **a** both eyes open > NREM ($n = 9$), **b** left eye open > NREM ($n = 11$), and **c** right eye open > NREM ($n = 12$) (group analysis using a mixed effects model, FLAME1, and multiple comparison correction using cluster-based FWE correction at a $Z = 2.3$, and $p < 0.05$). The functional maps were superimposed on the high-resolution anatomical data for the horizontal (left), axial (middle), and sagittal (right) axes of an ex vivo pigeon brain (in grayscale). The red to yellow scale shows areas with increased BOLD signal during eye-open conditions, whereas the blue to green scale shows areas with increased fMRI signal during NREM sleep. Brain regions with increased BOLD signal were largely restricted to the entopallium or surrounding visual areas contralateral to the open eye(s), whereas areas with increased fMRI signal during NREM sleep were largely restricted to the ventricular system. **d** To quantify the fMRI signal in these areas, we restricted the analysis to the entopallium (E, red area) and the IV ventricle and its cerebellar recess (IV and CR, blue areas). Estimated parameters from GLM analyzes for various conditions, including NREM, REM, left, right, and both eyes open, were extracted from voxels in E, IV and CR. **e** As with REM sleep, the BOLD signal in the entopallium was bilaterally elevated when both eyes were open;

when only one eye was open, the contralateral entopallium showed significantly stronger BOLD signal than the ipsilateral entopallium (two-tailed paired t-test, $t_{Left}(20) = 2.03$, $p = 0.039$; $t_{Right}(22) = 1.93$, $p = 0.046$, corrected for multiple comparison). **f** During all eye-open states and REM sleep, the fMRI signal in IV was decreased when compared to NREM sleep (two-tailed two-sample t-test, $t_{REM}(28) = 5.4$, $t_{Left}(24) = 3.6$, $t_{Right}(25) = 4.8$, $t_{Both}(22) = 3.1$, $p < 0.001$, corrected for multiple comparison). However, the estimated parameters were significantly lower during REM sleep when compared to the eye-open states (two-tailed paired t-test, $t_{Left}(24) = -2.7$, $t_{Right}(25) = -2.5$, $t_{Both}(22) = -3.2$, $p < 0.01$, corrected for multiple comparisons). **g** CSF signal in the CR demonstrated a decrease during both eye-open states and REM sleep, as opposed to NREM sleep (two-tailed two-sample t-test, $t_{REM}(28) = 5.2$, $t_{Left}(24) = 3.8$, $t_{Right}(25) = 4.8$, $t_{Both}(22) = 3.2$, $p < 0.001$, corrected for multiple comparison). Nonetheless, during REM sleep, the estimated parameters were notably lower when compared to the eye-open states (two-tailed paired t-test, $t_{Left}(24) = -3.3$, $t_{Right}(25) = -2.4$, $t_{Both}(22) = -2.5$, $p < 0.01$, corrected for multiple comparisons). Colored dots correspond to bird ID. Source data are provided as a Source Data file. L: left; R: right; A: anterior; P: posterior; ns.: not significant.

The cerebellar nuclei, including PCV, medial CbL, and ventral CbM, were also activated during REM sleep. They receive visual optic flow inputs from the collaterals of mossy fibers originating in the nucleus of the basal optic root[27]. REM sleep associated BOLD signals also covered those parts of the thalamic n. rotundus that are activated by looming objects on a stationary background[52]. These rotundal signals are then processed in the pallial visual tectofugal system and its associative visual areas[53], which were also active during REM sleep. Thus, it is tempting to speculate that our pigeons might have dreamed about diverse scenes of flying.

Despite the similarity of the neural activation patterns of birds and mammals during REM sleep, two differences deserve attention. First, we did not observe an activation of the thalamic lateral geniculate nucleus that in mammals is part of the ponto-geniculo-occipital wave which constitutes a prominent phasic event during REM sleep[38]. Instead, the n. rotundus was activated in pigeons - a structure that constitutes the thalamic relay of the tectofugal visual pathway. Second, the similar level of hippocampal formation activity during REM and

NREM sleep is inconsistent with some studies that found increased activation during REM sleep in humans[54], and a recent fMRI study in mice which shows lower activity during REM sleep, relative to NREM sleep[39].

Our findings indicate that, as in mammals, ventricular CSF flow increases during NREM sleep in birds, suggesting that waste clearance is a fundamental function of this sleep state. Although our findings indicate that functions linked to brain activation during REM sleep occur at the expense of ventricular CSF flow, REM sleep might still play a role in waste clearance. During the transition between NREM and REM sleep in rodents, the influx of blood into the brain[55,56] increases vascular diameter[57] and compresses the perivascular space through which CSF normally enters the brain[58]. This might restrict the entry of new CSF into the brain, and thereby account for the reduction in CSF flow outside the brain tissue in pigeons. However, at the same time, we propose that the increase in brain blood volume might squeeze the perivascular and extracellular spaces, thereby increasing flow through the brain tissue. The high levels of acetylcholine (Ach) and low levels of

norepinephrine (NE) occurring during REM sleep in mammals[59,60] might facilitate this process, as Ach causes dilation of the cerebral arteries and a corresponding influx of blood[61], and low levels of NE increase extracellular space and the movement of CSF through the brain[2]; however, it is unknown whether Ach and NE levels change in the same manner in birds. Regardless, according to this model, most of the flow should accompany the surge of blood at the onset of REM sleep. Consequently, by engaging in hundreds of short episodes of REM sleep, rather than fewer and longer episodes, as in mammals, birds might maximize waste removal. Indeed, this partitioning of REM sleep into shorter episodes could be an evolutionary strategy to enhance waste clearance in avian brains with high neuronal density.

## Methods

### Animal subjects

All experimental procedures were conducted under the National Institutes of Health Guidelines for the Care and Use of Laboratory Animals and were approved by the ethics committee of the State of North Rhine-Westphalia, Germany (Landesamt für Natur, Umwelt und Verbraucherschutz Nordrhein-Westfalen (LANUV), application number: Az.: 81-02.04.2021.A240). Fifteen adult domestic pigeons (*Columba livia*, Budapest highflyer variety; 7 females and 8 males, 2–3 years old, genetically sexed), obtained from a local breeder, were used in this study (Supplementary Table 2). Budapest pigeons were selected for this fMRI study because their small body fits well in the small-bore of the preclinical scanner resonator. In addition, their large eyes and transparent eyelids allow pupil size and eye movements to be monitored even when the eyelids are closed[21]. Before the experiments, birds were reared and housed in enriched colony aviaries, under 12 h:12 h light:dark photoperiod, at 21 °C. During the experiment, pigeons were individually housed in wire-mesh cages ($45 \times 45 \times 45\ cm^3$) under the same photoperiod with ad libitum access to water and food.

### Surgery and implantation

To prevent motion artifacts during the fMRI scans, all pigeons were implanted with an MR-compatible plastic pedestal. The protocol from our laboratory has already been published[20]. In summary, ketamine/xylazine (70% ketamine, 30% xylazine, 0.075 mL/100 g) was administered intramuscularly in the breast muscle to anesthetize the birds before implantation. In addition, a supplement of gas anesthesia (Isoflurane; Forane 100% (V/V), Mark 5, Medical Developments International, Abbott GmbH and Co. KG, Wiesbaden, Germany) was administered. After fixing the pigeon's head in a stereotactic apparatus[62], the skin and soft tissues around the skull were removed and four polyether ether ketone (PEEK) micro pan head screws were screwed into the skulls to attach the custom-made plastic pedestal. In addition to the plastic pedestal, six birds were implanted epidurally over the hyperpallium with wire electrodes soldered to a connector. Two electrodes were placed over the visual hyperpallium of both hemispheres and the third one was placed above the cerebellum as a reference electrode. The electrodes and connector were embedded in biocompatible skin glue, covered by a thin layer of dental cement. Finally, custom-made plastic pedestals and screws were embedded with dental cement (OMNIDENT, Rodgau, Germany) to increase the adhesive strength between the skull and pedestal. Following each surgery, analgesic (carprofen (Rimadyl), Zoetis Deutschland GmbH, Berlin, Germany 10 mg/kg) and antibiotic (Baytril, Bayer Vital GmbH, Leverkusen, Germany, 2.5 mg/kg) treatments were given every 12 h for at least 3 days. After a recovery period of 4–6 weeks with ad libitum access to water and food, the habituation training for head fixation started.

After recording the EEG signals, the electrodes were removed under isoflurane anesthesia (2–3% in one LPM oxygen) before running any fMRI scans to avoid interfering with fMRI signals.

### Habituation procedure

To habituate the pigeons to the head fixation system (Fig. 1), they were gradually acclimated to the restrainer and acoustic noise of the magnet in a mock scanner using a well-established procedure in our laboratory[20,63,64]. This procedure helps to reduce the stress associated with head fixation and minimizes body motion artifacts. As a result of the procedure, the birds fell asleep after a few minutes inside the scanner. In brief, the habituation protocol consists of three main steps: (i) to habituate the animals to the experimental environment and the restrainer, pigeons were wrapped in a cloth jacket to prevent wing and leg movements and then placed in the restrainer inside the mock scanner in a dark room. (ii) to habituate the animals to the head fixation, animals were fixed to the holding device via the implanted MR-compatible plastic pedestal. The head fixation started with 10 min on the first day and was prolonged to 100 min in 12 days; (iii) to habituate pigeons to the scanner noise, a recorded single-shot Rapid Imaging with Refocused Echoes (RARE) sequence sound was replayed during head fixation with gradually increasing sound pressure from 45 to 90 dB at one meter distance. After 18 days of habituation, all birds engaged in NREM and REM sleep in the training setup and proceeded to the next phase of obtaining the fMRI recordings.

### EEG acquisition

Infrared (IR) illumination and IR-sensitive video cameras were used to monitor the sleep behavior. Video recordings were done at 30 fps. The EEG signal of the six birds implanted with electrophysiology electrodes was additionally recorded during the training phase at 256 Hz using the same recording system (Oneiros), as in our previous studies of Budapest pigeons[21,65,66].

### fMRI acquisition

Prior to the main experiment, the EEG electrodes were removed from the 6 implanted birds, and the photoperiod was inverted to facilitate sleep recordings during the daytime. As pigeons, like diurnal mammals, engage more frequently in REM sleep later in the night[67], the fMRI recordings were performed in the second part of the birds' subjective night.

All MRI data collection was carried out in a Bruker BioSpec 7 Tesla scanner (horizontal bore, 70/30 USR, Avance III electronic, Germany) using Paravision 6.0 software. A quadrature birdcage resonator (82 mm ID) was used for RF transmission and a single-loop receiver surface coil (20 mm ID) was used for resting-state and anatomical data collection. By positioning the ring surface coil around the head, it was possible to reduce artifacts due to body movements. As changes in respiration may affect BOLD signals and brain connectivity maps, the respiration waveform was measured using a small pneumatic pillow placed under the pigeon's chest muscles (Small Animal Instruments, Inc. Model 1025 T monitoring and gating system) during all resting-state measurements. The facial behavior of the animals was simultaneously recorded using an MRI-compatible video camera (12M-i, MRC Systems, Heidelberg, Germany; B/W) with an incorporated LED light (IR) (outside the pigeon's visible spectrum). To facilitate detection of changes in pupil size and eye position, a small mirror was placed next to each eye (Fig. 1a). The videos were acquired at 30 fps. All recordings including respiration signals, videos, and resting-state data were synchronized using a TTL signal.

To localize the correct position of the pigeon brain within the scanner bore, a series of scout images were measured at the beginning of each scanning session. Three runs (horizontal, coronal, and sagittal) were acquired using multi-slice rapid acquisition (RARE) with the following parameters: TR = 4 s, TEeff = 40.37 ms, RARE factor = 8, no average, acquisition matrix = $128 \times 128$, FOV = $32 \times 32$ mm, spatial resolution = $0.25 \times 0.25\ mm^2$, slice thickness = 1 mm, number of slices = 20 horizontal, 17 sagittal, and 15 coronal. These images were used to orient 11 coronal slices to cover the entire telencephalon, optic lobes, midbrain, IV ventricle, and cerebellum.

Resting-State fMRI (rs-fMRI) data were acquired using a single-shot multi-slice RARE sequence adapted from Behroozi et al.[20,64], with the following parameters: TR = 4000 ms, TEeff = 41.58 ms, partial Fourier transform accelerator = 1.53, encoding matrix = 64 × 42, acquisition matrix = 64 × 64, FOV = 30 × 30 mm$^2$, in-plane spatial resolution = 0.47 × 0.47 mm$^2$, radio-frequency pulse flip angles for excitation and refocusing = 90°/180°, slice thickness = 1 mm, no slice distance, slice order = interleaved, excitation and refocusing pulse form = scanner vendor gauss512, receiver bandwidth = 50,000 Hz. To saturate the signals from the eyes to avoid brain image corruption due to eye movements, two saturation slices were positioned manually over the eyes. Each run of the rsfMRI recordings included 1000 to 1450 volumes. To check the reproducibility and stability of the results, rsfMRI data of all animals were recorded twice on different days.

High-resolution T2-weighted anatomical images were acquired using a RARE sequence for better spatial normalization. Scan parameters were as follows: TR = 2000 ms, TEeff = 50.72 ms, RARE factor = 16, number of averages = 1, FOV = 25 × 25 × 15 mm$^3$, matrix size = 128 × 128 × 64, spatial resolution = 0.2 × 0.2 × 0.23 mm$^3$. The total scanning time was 17 min.

To represent the results in high-resolution anatomical images, following the final MRI data acquisition, one of the animals was deeply anesthetized with equithesin and transcranial perfused with a phosphate-buffered saline solution (PBS, 0.12 M), followed by a mixture of paraformaldehyde (PFA 4%) and Dotarem® (1%). High-resolution 3D anatomical data was collected using a T2*-weighted sequence using a FLASH sequence with a flip angle of 15°, a spectral bandwidth of 50 kHz, 64 averages, a TR of 9.47 ms, a TE of 4.738 ms, and a scan repetition time of 5.000 ms. The images have a field of view of 25 × 25 × 20 mm$^3$ and an isotropic spatial resolution of 0.05 mm in all three directions. The recording time was ~ 20 h.

## Awake condition

As our birds closed their eyes and fell asleep almost immediately after head fixation, we had to wake them up. Therefore, we played loud auditory stimuli during the training EEG and fMRI acquisition. Auditory stimuli included a piece of classical music (first 8 s of Brandenburg Concerto No. 4 of Johann Sebastian Bach) and two random chord stimuli centered at 1000 Hz (500–1500 Hz) and 3000 Hz (1500–4500 Hz). The auditory stimuli were presented using an MR-compatible speaker (SoundCraft) which was placed at a distance of 8 cm from the tip of the bill. In total, 45 trials were played during the EEG and fMRI recordings. Each trial consisted of 8 s auditory stimuli followed by 52 s of silence. We digitized all stimuli at 44.1 kHz. The maximum sound pressure level (SPL) was approximately 90 dB (measured at a 1 cm distance from the speaker).

## Sleep state scoring

Unilateral and bilateral eye opening, and NREM and REM sleep states were manually scored using the video recordings and 1 s epochs. To facillitate the scoring process, actimetry signals were calculated for the eyes, irises, and bill region and visualized together with the video recordings. NREM sleep was characterized by bilateral eye closure, stable breathing (as visible in the videos and bill actimetry signals), immobility of the eyes, and absence of bill movements other than those related to breathing. REM sleep was characterized by bilateral eye closure with movements of the eyes, bill, and/or the collapse of head feathers held erect during preceding NREM sleep. As shown previously[21], rapid constrictions and dilations of the iris are closely associated with REM sleep. Thus, rapid iris movements were also used to identify bouts of REM sleep and their absence to confirm NREM sleep. To further reduce variability and avoid including potential transitional states in the analysis, two seconds at state change and all ambiguous epochs were discarded. Left and right eye opening was scored when the respective eye was open more than two thirds, while the other one was completely closed. Bilateral eye opening was defined by both eyes being more than two thirds open. In all cases, a given state was attributed to an epoch only when it occupied 100% of its duration.

## EEG data processing

The EEG data was analyzed using custom Matlab (v2020b) scripts. First, a time-frequency analysis using a multitaper method[68] was performed for the complete recordings session using the following parameters: $F_{min}$ = 0, $F_{max}$ = 40 Hz, window size = 4 s, step = 50 ms, band width = 1 Hz, taper number = 3, pad = 1. A time window of 40 s centered on the sound onset was exctracted for each stimulation trial. The signals were averaged across trials to obtain an average time-frequency representation. To calculate the average power spectrum associated with the auditory stimulation, the signal was averaged across the 8 s of stimulation. A similar averaging was done for a 8-s time window prior to the stimulation. The same parameters were used to calculate the average power spectrum for behaviorally scored NREM and REM sleep in head-fixed pigeons inside the mock scanner (out of MR machine).

## BOLD fMRI data processing

All fMRI data processing was performed using tools from the FMRIB Software Library (https://fsl.fmrib.ox.ac.uk/fsl/fslwiki/FSL, version 5.0.9), the Analysis of Functional NeuroImages (AFNI, version 20.0.09 https://afni.nimh.nih.gov/), and Advanced Normalization Tools (ANTs, http://stnava.github.io/ANTs/, version 2.1 1) software. A standard pipeline was used to pre-process the rsfMRI data. Prior to any preprocessing steps, the DICOM images were converted to 4D NIFTI data (using the *dcm2nii* function), and the voxel size was upscaled by a factor of 10 (using the *3drefit* function of AFNI). The censoring of high motion frames was applied to estimate the amount of head movement within each resting-state time series[69]. The threshold for framewise displacement (i.e., volume to volume movement) was set as 0.9 mm (less than 20% of voxel size). The results indicated that only 25 volumes of all 29 rs-fMRI sessions had an FD-value higher than 0.9 mm (after upscaling voxel size by a factor of 10). Later, the following standard data pre-processing steps were applied: (i) motion correction using MCFLIRT[70] (FSL's intra-modal motion correction tool), (ii) slice time correction (interleaved acquisitions, using the *slicetimer* function), (iii) skull stripping of functional data (using a BET[71] and manual cleaning), (iv) spatial smoothing (using *3dBlurInMask* function with FWHM of 8 mm, after upscaling), (v) global intensity normalization by a single multiplicative factor for each scan run (for group analysis), (vi) high-pass temporal filtering (using the *3dTproject* function, with cutoff at 0.01 Hz). The first 5 and last 5 volumes were discarded to ensure longitudinal magnetization reached a steady state and also to avoid the edge effect of temporal filtering. To consider the respiratory-related artifacts, voxel-wise regressors for physiological noise based on respiratory signals were generated using the PNM tool in FSL[72] by calculating the respiratory phases relative to each volume and slice in the rs-fMRI signals.

We then co-registered each rs-fMRI data to the corresponding T2-weighted anatomical images using affine linear registration (12 degrees of freedom). A population-based template was generated using *antsMultivariateTemplateConstruction.sh* script[73]. After analyzing individual subjects, the results were normalized to the population-based template using FMRIB's Nonlinear Image Registration Tool (FNIRT)[74] for group analysis. For visualizing the results, the group results were non-linearly warped to the high-resolution post-mortem anatomical image. 3D MRI images were visualized using the MANGO software (http://ric.uthscsa.edu/mango/, version 4.1).

## ROI definition

The fourth ventricle (IV), its extension into the cerebellum, namely the cerebellar recess (CR), and the gray matter mask were defined anatomically in the high-resolution anatomical image (from the perfused animal). The brightest voxels in the anatomical image were selected to

identify the position of IV and CR. The CSF masks were selected from the bottom slices of the rs-fMRI data to ensure high sensitivity of the 2D rs-RARE images to the CSF through-slice inflow effect as mentioned previously[9,29]. Since there is no blood in the CSF regions, the fMRI signal changes in these regions are mainly due to variations of the inflow of fresh spins into the imaging volume[9,75]. This inflow effect size is much stronger at the edge of the imaging volume because the fresh spins arriving have not yet experienced magnetization saturation due to the radiofrequency pulses[75]. We, thus, extracted the CSF signals from the voxels within lower slices[9,29] to estimate an index of the change in CSF flow over time.

ROIs for the BOLD signal in the gray matter were defined by two spheres with a diameter of 6.5 mm on both hemispheres (centered at the Entopalium) to include most of the telencephalon. In addition, the hippocampal formation mask was defined based on pigeon MRI atlas[76]. We transformed the selected masks from the high-resolution anatomical space to the functional space of each pigeon using the reversed concatenated transformation matrix from the previous steps. However, the masks in the functional space were also visually inspected to ensure their correct position. All ROI analyses and value extraction were done within the fMRI acquisition space of each pigeon to avoid any spatial blurring of the original fMRI signal during transforming into the template space.

### Statistics and reproducibility

All linear mixed model analyses were performed in R using the *lmer* package. The other statistical tests were performed in MATLAB (v2020b). A two tailed paired t-test was used to compare the estimated parameters from different ROIs and conditions. The level of statistical significance was set to $p < 0.05$. The number of NREM and REM sleep bouts in the two analyzed sessions was tested using a paired Student's t-test. To assess whether the sound stimulation (to awaken the birds) influenced the EEG activity in the hyperpallium (a primary visual area), we calculated the average power in the delta band (0.5-4 Hz) 20 s before and during the stimulation (Supplementary Fig. 9). The effect of the stimulation was calculated using linear mixed effect models, with the average delta power before, or during the stimulation as the response variable, the condition (stimulation vs non-stimulation) as a predictor, and the bird identity as a random effect.

Whole-brain statistical analysis was carried out using FEAT (FMRI Expert Analysis Tool, a part of FSL) with high-pass temporal filtering (cut-off 100 s) and pre-whitening using FILM[72] on individual pigeons. The general linear model (GLM) included a regressor for each of the left eye open, right eye open, both eyes open, NREM sleep, and REM sleep bouts, and their temporal derivatives. In addition to the five explanatory variables, the PNM regressors (to account for the effects of physiological noise in the BOLD signal), six scan-to-scan estimated motion parameters, and frames that exceeded a threshold of 0.9 mm FD (to account for any residual effects of animal movement) were added as confounds in the GLM model. The experimental regressors of interest were used to create the following brain activity contrasts for the different sleep stages: positive and negative main effects of REM versus NREM sleep, as well as left eye open versus NREM, right eye open versus NREM, and both eyes open versus NREM sleep.

The birds were recorded for three sessions, each on independent days. The first session served for habituating the bird to the magnetic field, while the last two sessions were analyzed. Due to low amounts of NREM and REM sleep in the last session of one pigeon, only the second session was included in the analysis. Each session was analyzed independently. A two-sample paired t-test (implanted in FSL) was used to reveal potential changes in activity on the first and second days. Since there was no significant difference between the two sessions (Supplementary Fig. 6 and Supplementary Fig. 7), within-subject effects (across the two days) were then established in second-level analysis (fixed effects) to average the two sessions from each pigeon. We

looked for sex differences (two-sample unpaired t-test). The results showed no significant differences between male and female pigeons. For this reason, we combined data from males and females for subsequent analyses.

For the REM versus NREM group analysis, contrasts of interest from second-level analyses were then taken into the higher-level analysis using the nonparametric FSL's *Randomise* function[77]. Statistical significance was assessed using permutation testing with 5,000 permutations, Threshold-Free Cluster Enhancement (TFCE), and a family-wise error (FWE) corrected cluster p-value of $p = 0.05$. For the open eye conditions group analysis, contrasts of interest from second-level analyses were then taken to the higher-level analysis using the mixed-effect model (FLAME1). Higher-level results were thresholded using activation levels determined by $Z > 2.3$ ($p < 0.01$) and FWE cluster significance threshold of $p = 0.05$.

### GM and CSF signal analysis

In addition to the mentioned preprocessing steps, the "*film_gls*" and "*fsl_regfilt*" functions were used to regress out the voxelwise explanatory variables of respiration signal and estimated motion parameters (3 translations and 3 rotations), respectively. To have equal fluctuation amplitudes, the fMRI signals were normalized to Z-score at each voxel. For each resting-state session, the mean fMRI signals from each mask were extracted. For the ROI signal analysis of GM and CSF as a function of time and sleep state, we selected NREM/REM bouts whose onset were a minimum of 12 s away from NREM/REM bout offsets to minimize any possible effect of the adjacent NREM and REM bouts on each other.

### Reporting summary

Further information on research design is available in the Nature Portfolio Reporting Summary linked to this article.

## Data availability

The fMRI data generated in this study are publicly available in BIDS format and have been deposited in OpenNeuro (Accession number: ds004465) as functional MRI of sleeping pigeon[78]. The EEG data are available at (https://doi.org/10.17617/3.JFRZHS)[79]. All information regarding the timing of REM and NREM, obtained from videos, has already been shared within the fMRI data. The source data underlying Figs. 1, 4, 5 and Supplementary Figs. 1, 2, 5, 7, and 10 are provided as a Source Data file. Source data are provided with this paper.

## Code availability

FSL (https://fsl.fmrib.ox.ac.uk/fsl/fslwiki/FSL, version 5.0.9), the Analysis of Functional NeuroImages (AFNI, version 20.0.09 https://afni.nimh.nih.gov/), and Advanced Normalization Tools (ANTs, http://stnava.github.io/ANTs/) software and MATLAB (v2020b, MathWorks, USA) were used to process fMRI and EEG data. Codes for fMRI data processing are available at GitHub (https://doi.org/10.5281/zenodo.7851602)[80]. Code for EEG data processing are available as m files at https://doi.org/10.17617/3.JFRZHS.

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

## Acknowledgements

This work was supported by Max Planck Society (G.U., L.B., N.C.R.), grants from Deutsche Forschungsgemeinschaft (DFG, German Research Foundation)—Projektnummer 316803389—SFB 1280 (O.G.), and AVIAN MIND, ERC-2020-ADG, LS5, GA No. 101021354 (O.G.). We are grateful to the animal care staff at the Max Planck Institute for Biological Intelligence and Ruhr University Bochum for taking care of the birds. We acknowldege support by the Open Access Publication Funds of theRuhr-Universität Bochum. Open Access funding enabled and organized by Projekt DEAL

## Author contributions

The lead contributors are listed first, in alphabetical order, and marked with *. The other contributors are listed in alphabetical order. Conceptualization: N.C.R.*, G.U.*. Methodology: M.B.*, G.U.*, P.-A.L., X.H. Investigation: M.B.*, G.U.*, L.B. Visualization: M.B.*, G.U., L.B. Funding acquisition: O.G.*, N.C.R.*. Project administration: G.U.*, M.B.*, O.G., N.C.R. Supervision: O.G.*, N.C.R.*. Writing—original draft: M.B.*, O.G.*, N.C.R.*, G.U.*. Writing—review & editing: M.B.*, O.G.*, N.C.R.*, G.U.*, L.B., P.-A.L., X.H.

## Funding

## Competing interests

The authors declare no competing interests.
