## [Peer Review File · Nature Communications]

REVIEWER COMMENTS

Reviewer #1 (Remarks to the Author):

This is a very interesting study that examines brain activity and CSF flow using fMRI in sleeping pigeons. As in mammals, periods of REM sleep, which are very brief in pigeons, are accompanied by a paradoxical increase in brain activity and reduction in CSF flow. They conclude that, as in mammals, non-REM sleep is important for clearing toxic proteins in the brain through the glymphatic system. Based on the brain activity they also speculate on what pigeons are dreaming about. In this regard, based on the widespread activity in telencephalic visual areas and an optic flow region in the cerebellum, they suggest that pigeons are dreaming about avoiding obstacles during flight. A comment from one of the authors that was inadvertently left in the margin of the manuscript betrayed the authors concerns about referring to "dreaming" during REM sleep, perhaps as too anthropomorphic. In my opinion, they are well justified in such speculation.

This paper was a pleasure to read and I am sure will be of interest to the broad readership of Nature Communications. With the supplemental material and detailed analyses I am very convinced about the scientific rigor employed in this study. I have a few minor comments.

page 4, line 21. - "two of the recorded fMRI sleep sessions were analysed". I was very confused by this sentence and think the authors need to provide clarity. Does this mean that of all the instances of periods of REM and non-REM sleep they only analysed two instances? Or is the "two" referring to the two states of sleep? Please provide some clarification.

page 5, line 28. - I think it is wrong to use the wording "statistically similar". The test is to see if there is a difference at $\alpha = 0.5$, not if there is similarity at $\alpha = 0.95$. I would suggest stating that the number of NREM and REM bouts was "not significantly different".

Fig 2. In the text there is no mention of the intense BOLD section in the most caudal coronal section (5th row third column). In the accompanying figure S5, this is the only coronal section where the nuclei are not delineated. It looks to me that this activity is in the cerebellovestibular process (pcv) and perhaps adjacent regions in the ventral part medial cerebellar nucleus (CbM), the medial part of the lateral cerebellar nucleus (CbL), and perhaps the dorsal part of the superior vestibular nucleus. These areas (pcv, medial CbL and ventral CbM) receive visual optic flow inputs from collaterals of mossy fibres originating in the nucleus of the basal optic root (Wylie et al. JCN, 1997). This lends support to the author's suggestion that the pigeons are dreaming about flying.

page 10 line 5. It is stated here that the nucleus rotundas is activated by "optic flow". This is incorrect. The confusion may be that many neurons are responsive to looming objects on collision course, but Wang et al. emphasized that these were looming objects on a stationary background, but not looming surfaces.

page 10 lines 23-30. I find these statements to be rather speculative, and not supported by scientific findings. Can the authors reference any studies that suggest that the increase in blood flow does aid in waste removal?

page 13 line 7. The reference to Wylie (2013) is incorrectly listed as volume 0, but is actually volume 7. They refer to this study in regards to the activation of folium VI being concerned with optic flow. A more appropriate reference might be by Wylie et al. (2018, *Frontiers in Neuroscience* 12:223) where the authors directly implicate the oculomotor cerebellum in flight through visually cluttered environments.

page 13 line 2; What do the authors mean by the birds were "ready" to be restrained? Could they offer a more descriptive operational definition? Was it simply that the birds readily fell asleep? Did some pigeons fail to habituate.

page 16, lines 10-15. How long to the birds remain awake after the presentation of the auditory stimulus?

Figure captions for S4 and S6. I am confused. It is stated that green is day 1, and yellow is day 2. Is this correct? Or is it that day 1 in BLUE and day 2 is yellow and green represents the overlap?

Reviewer #2 (Remarks to the Author):

In this manuscript, the authors report two original findings both of which have significant relevance to the field of sleep. First, these kinds of explorations of bird sleep are fundamental to understanding human sleep. As the authors outline, birds, like mammals, express both REM and NREM sleep and these sub states share many electrophysiological and behavioral features. Despite the similarities, the states are not identical and sleep in birds evolved independently from mammalian sleep. This independent evolution suggests that sleep, and this two-state expression of it, must serve some fundamental

function for the nervous system; if it weren't essential, evolution would have dispensed with. Instead, it seems, nature made only subtle modifications. In light of this, exploring the similarities and differences between mammalian and avian sleep, is an ideal way to reveal fundamental functions of NREM and REM sleep. Essentially, this type of comparative work allows us to see which features of sleep nature kept along the journey of evolution and to explore why they are so important.

The paper focuses on the role of sleep in glymphatic clearance in an avian model. The recently discovered glymphatic system is a CSF circulation pathway responsible for metabolic waste clearance; in mammals CSF circulation has been shown to increase during sleep in mammals and this increase is associated with clearance of toxins including, but not limited to, excess glutamate, lactate, and amyloid-beta. Although the vast majority of glymphatic clearance in mammals is thought to occur during sleep, recent data also suggests that the process may also be modulated by the circadian system. Ultimately, the details of how metabolic waste products are cleared from the brain and the specific role mammalian sleep plays in this process is not entirely clear. A detailed understanding of sleep's role in this process has far-reaching importance, in humans, for physical and mental illnesses. By taking a comparative view of glymphatic function in sleeping birds, the authors provide evidence supporting a central role of NREM sleep in glymphatic activation. They also show something that has not been shown in a mammalian model; REM sleep does not support glymphatic clearance. The fact that this question has never been addressed in a mammalian system represents a major gap in the literature.

The resting state BOLD data is also significant. Here the authors demonstrate that spontaneous brain activity in birds is shaped in a state specific way that largely recapitulates mammalian sleep (including humans). These data, and the fact that resting state data during sleep can be captured in birds without the use of anesthesia, have implications for our understanding the evolutionary principles of consciousness. Although a small number of very recent studies have successfully used functional neuroimaging in behaving birds, resting state data in awake animals is exceedingly difficult and, except in humans, has been almost exclusively collected under anesthesia—which significantly limits understanding of the brain's functional activation and architecture. Using the methodology described here, the authors can further explore the dynamic structure of the avian brain and its reconfiguration as a function of distinct states of consciousness (i.e. NREM unconsciousness vs REM consciousness) in a comparative way. In my view, the BOLD data shown here represent another step forward in the path towards dissolving our belief in mammalian exceptionalism.

Although my view of the paper is overwhelmingly positive, below are minor questions related to methodology and interpretation.

- 1) As the authors describe, sleep structure and regulation in birds is quite similar to mammalian sleep, although REM and NREM states, as well as intermediate sleep (IS; dozy) bouts, are shorter and state transitions more frequent. My understanding of sleep in captive birds, informed in part by the careful work done by the last author (NR), is that when birds are not actively engaged in bird stuff (technical

term encompassing the entire behavioral repertoire of the captive bird: eating/drinking, hopping/flapping, singing/calling, scratching/beak scraping) they are sleeping. One way to keep them from sleeping is to introduce unfamiliar sensory stimuli— the unusual musical selections used here would seem adequate. Given that the authors seem to have very clear data on eye closure, effectively tracking any unihemispheric sleep, why are the analyses largely restricted to REM vs NREM comparisons? They do compare to 'baseline' but they do not really make substantive comparisons with wake? If they plan to publish a distinct manuscript focused on glymphatic clearance as a function of behavioral state (sleep v wake) that should be mentioned. However, instead it seems that they are unable to clearly delineate NREM from wake. If so, what was the cause of this? Were bouts and transitions impacted by the inversion of the sleep period from night to day (i.e, was behavior, temporal distribution and bout duration, number of unihemispheric event unlike typical sleep)? Or are wake bouts generally too short during the sleep period to make meaningful comparisons with wake possible?

2) Given the brief duration of REM episodes and the frequent state transitions in birds, why was a TR of 4 seconds chosen to record whole-brain BOLD responses? Was there a reason for not choosing shorter (i.e 2 sec TR) sequences which are often used in task-related functional imaging paradigms in humans? Does this long TR have implications for the regressors associated with physiological noise (HR, respiration) in REM and NREM and their removal in the GLM?

3) The authors say that they “characterized the species-specific hemodynamic response function”. Do they mean they estimated activation parameters-lag time from state of interest, duration of bold signal, time to peak signal, max signal strength, from these animals? Or from existing literature? Is the change in maximum BOLD signal strength similar across brain regions and across behavioral state? If not, might this impact their ability to detect BOLD differences, for example, in the hippocampus in NREM vs REM? It is certainly conceivable that birds process memory differently than mammals, but could there be another explanation for the failure to see a mammalian-like hippocampal activation in REM.

4) What was the resolution of sleep state scoring? I assume it was continuous (not epoch based?)

5) Noradrenergic cell groups in the pontine brainstem are conserved across fish, amphibians, reptiles, birds, and mammals, and play universal roles in modulating behavioral state, attention and stress. Although far less is known about the levels of brain norepinephrine (NE) during avian REM sleep, the conservation of the ascending arousal system would suggest that forebrain NE is similarly modulated in birds and mammals during NREM and REM sleep such that REM sleep is a state likely free of NE. Data in mammals suggests that NE plays a central role in glymphatic clearance. During mammalian sleep, the story goes, the decrease in NE levels causes the expansion of the extracellular space, decreasing resistance and increasing the rate of glymphatic clearance. Can the authors comment on how this story fits, or doesn't fit, with their data? Is there insufficient information on NE regulation in birds to conclude that it is a state void of NE? Differences in glymphatic clearance during NREM vs REM have not yet been

demonstrated in mammals, and this seems remarkable to me given the imputed importance of NE in the process.

6) As you will have guessed from my summary of the manuscript above, I find the BOLD data fascinating. The authors compellingly demonstrate that REM sleep is associated with activation in sensory motor and limbic regions in a mammalian-like way. When reflecting on these data in the Discussion they say, avoiding collisions with other objects.”

I assume these are captive birds? Existing data certainly supports the idea that birds are capable of higher-level consciousness-- as does common sense. Anyone who has ever lived near a red-winged black bird knows they exhibit a level of consciousness indistinguishable from that of the human bully; a show of weakness is grist for the mill. A natural extension of this higher ordered thinking is that birds have the ability to recapitulate, during sleep, previously lived experience. However, it's still hard to imagine that a bird might spontaneously imagine the experience of flight, including obstacle avoidance, and relive that experience during REM sleep.

Reviewer #3 (Remarks to the Author):

This unusual and interesting paper examines CSF dynamics in the sleeping pigeon using fMRI. How sleep modulates brain fluid physiology is a topic of high interest, and very little prior work exists on REM sleep. This paper performed fMRI of sleeping pigeons and concludes that pigeon NREM sleep involves increased CSF flow (suggesting birds exhibit similar dynamics to humans) and that REM sleep involves decreased CSF and widespread increases in BOLD. The analysis methods are overall sensible, with some issues needing clarification. This work provides a novel report of sleep affecting CSF in birds, which are evolutionarily distant from humans, as well as one of the first reports of CSF imaging in REM sleep.

Comments:

- The sleep scoring was done manually using only the video recordings of the pigeons' pupil and head movements without any measurement of simultaneous EEG activity to confirm the sleep scores. It is true that EEG-fMRI is challenging, however, the authors had simultaneous EEG and pupil access outside the scanner. It would be useful to know how accurate this approach is, which they could either validate in their own data, or specify from prior work. For example, it seems very plausible that birds are sometimes awake with their eyes closed, which means part of the NREM bouts would be wake bouts.

- The Introduction states that they combine fMRI, EEG, and pupillometry, but EEG was not actually acquired during the fMRI measurements. It seems that EEG was only acquired during an auditory stimulus, and the relevance of the finding that auditory stimuli elicit EEG responses in the pigeon is not clear. Could this be clarified? Why is a sound EEG experiment present in the supplement fig 8? It seems there is a concern that birds sometimes sleep with their eyes open and this would lead to mislabeling of the bouts. However since EEG and auditory stimuli were not used in the main experiment, this control measure does not seem to help address the issue, and rather it seems to suggest that the wake bouts may be contaminated with sleep when acquired in the fMRI scanner?

-Is it typical for pigeons to have as much REM as NREM or is this because the recordings were performed towards the end of the bird's perceived night?

- How long were bouts? This information is important for interpreting the fMRI analysis, as the duration of bouts in the GLM will strongly affect its performance.

- In Fig 4b, the red and blue bars mark bouts of NREM and REM, but they have large gaps between them. What are the time gaps between bouts? Is it Wake data? If so, why would the pigeon wake up for prolonged periods between each bout? If not, why were those data not also modeled?

- Fig. 4 is very hard to read; I'd recommend putting line plots on a white background. The lines representing model fit of GM and model fit of CSF are not easy to see. Why is there a red dashed line in 4C? The legends are difficult to read.

- What do different colored circles represent in fig S9? It is unclear why there are only four of them.

- "Thus, it is tempting to speculate that our pigeons might have dreamed about scenes of flying while avoiding collisions with other objects." this is indeed extremely speculative.

- It is not clear what the three colour bars in Supp Fig 3 represent. The legend states that Day 1 is green, and there is green in the image, but there is no green colour bar. The same issue is present in Supp Fig 5.

Minor

- supp fig 1: typo, 'Fremwise' should be 'frameworkise'

Reviewer #4 (Remarks to the Author):

In this report, the authors describe different brain activations during REM and no-REM sleep states in pigeons, using fMRI, with widespread activations in the grey matter regions during REM and 'activations' in the ventricles during NREM. Technically, the experiment is impressive and the results are interesting in themselves. However, their interpretation is highly speculative, which leaves me with mixed feelings regarding the suitability of the manuscript to be published in this journal.

The goal of the study is too vague: the abstract and the introduction seem to suggest that the original goal was to test specific hypotheses related waste clearance but this aspect is not manipulated and towards the end of the introduction, the authors states was their goal was to 'probe the brain mechanisms underlying REM and NREM sleeps'. If the original goal was descriptive rather than hypothesis-driven, the authors should be transparent about it and the interpretation of fMRI activations in the ventricles in terms of CSF flow and waste clearance should be restricted to the discussion. Alternatively, if the authors' intentions were to measure waste clearance/CSF flow by measuring fMRI signal in the ventricles, I would have expected some validation of the MRI signal, since it is unusual (fMRI signal in the ventricles are usually interpreted as artefacts and ignored).

The authors need to do more to convince the reader that the birds are sleeping in the scanner, despite the noise induced by the fMRI sequence. What was the intensity of the fMRI sequence? An audio file of the auditory stimuli used to 'wake up' the birds with the background noise of the scanner would be helpful. In human MRI experiments, it is common to see participants closing their eyes in the scanner if they do not have any task to do. However, only a minority of them report to have fallen asleep. Do the EEG data allow to exclude the possibility that the animals were in an 'awake with eye closed' state?

I cannot make sense of the contrasts REM vs 'Baseline' and NREM vs 'baseline', since 'baseline' refers to a mixture of various states. The comparison of fMRI activations between REM vs awake states (as well as NREM vs. Awake) would be more useful (even if the auditory stimulation would influence the results).

In humans, it is well established that activation of one brain region does not correspond to one single (perceptual or cognitive) process and therefore that the location of activations in the brain is not sufficient to infer the underlying experience of the participants. The interpretation of the results made by the authors is therefore highly speculative. Eye movements are also known to induce widespread brain activations in humans. I am therefore surprised that the authors did not discuss the possibility that

(at least some of) their results observed during REM sleep simply reflect eye movements rather than 'perceptual' experience.

REVIEWER COMMENTS

Reviewer #1 (Remarks to the Author):

This is a very interesting study that examines brain activity and CSF flow using fMRI in sleeping pigeons. As in mammals, periods of REM sleep, which are very brief in pigeons, are accompanied by a paradoxical increase in brain activity and reduction in CSF flow. They conclude that, as in mammals, non-REM sleep is important for clearing toxic proteins in the brain through the glymphatic system. Based on the brain activity they also speculate on what pigeons are dreaming about. In this regard, based on the widespread activity in telencephalic visual areas and an optic flow region in the cerebellum, they suggest that pigeons are dreaming about avoiding obstacles during flight. A comment from one of the authors that was inadvertently left in the margin of the manuscript betrayed the authors concerns about referring to "dreaming" during REM sleep, perhaps as too anthropomorphic. In my opinion, they are well justified in such speculation.

This paper was a pleasure to read and I am sure will be of interest to the broad readership of Nature Communications. With the supplemental material and detailed analyses I am very convinced about the scientific rigor employed in this study. I have a few minor comments.

We thank the reviewer for their positive overview and the minor comments listed below. Yes, as we inadvertently revealed, we struggled with how to address the question of whether the wide-spread brain activation occurring during REM sleep indicates that the pigeons were dreaming. Strictly speaking, we do not know whether pigeons dream, because they can not report their previous thoughts upon awakening; this is the only way that researchers know that humans dream. Consequently, stating unequivocally that pigeons dream seems unwarranted. Nonetheless, ignoring the possibility that the wide-spread brain activation occurring during REM sleep, particularly in visual regions, might be accompanied by dreaming also seems unwarranted. Therefore, we chose to address this important and fascinating issue by discussing how the networks activated might relate to the content of potential dreams.

We are not alone in relating wake-like brain activity and behavior occurring during sleep in non-human animals to potential dreams. Indeed, the replay of neuronal sequences occurring during wakefulness in the hippocampus of rats during REM sleep (Louie and Wilson Neuron, 2001) has been interpreted as potential signs of dreaming, as have periodic bouts of retinal movements in jumping spiders (Rößler et al. PNAS, 2022) and eye movements and changes in skin pattern, brightness and texture in cephalopods (Medeiros et al. iScience, 2021). In this regard, we also think that our findings will have a large impact, as there is a growing interest in the question of whether animals dream (Manger and Siegel J Comp Neurol, 2020; Malinowski et al. Conscious Cogn, 2021). Finally, as birds can be readily trained to behaviorally report on many cognitive processes, including consciousness (Nieder et al. Science, 2020), it is conceivable that our study will form the foundation for future experiments aimed at training pigeons to report their previous visual experiences upon awakening. For these reasons, and given that we clearly state that our comments on potential dreaming is speculative, we would prefer to keep this material in the discussion.

1- page 4, line 21. - "two of the recorded fMRI sleep sessions were analysed". I was very confused by this sentence and think the authors need to provide clarity. Does this mean that of all the instances of periods of REM and non-REM sleep they only analysed two instances? Or is the "two" referring to the two states of sleep? Please provide some clarification.

Each individual underwent three recording sessions, each of them performed on different days. Each recording session was 100 minutes long. During this time, the birds cycled through multiple NREM and REM sleep episodes. The first session served to get the animals used to the real magnetic field. The last two sessions were analyzed. A separate analysis of the two sessions demonstrated the reproducibility of our results. Given that the results from the two sessions were similar (Fig. S6), the data from these two sessions were combined and analyzed together as described in the Methods in order to increase overall sensitivity of the analysis.

To clarify this point, we changed the text as follows:

Main text: "After completing the habituation phase, the last two sessions were independently analyzed, then combined using the fixed effect modelling (Methods)"

Methods: "The birds were recorded for three sessions, each on independent days. The first session served for habituating the bird to the magnetic field, while the last two sessions were analyzed. Due to low amounts of NREM and REM sleep in the second session of one pigeon, only the last session was included in the analysis."

2- page 5, line 28. - I think it is wrong to use the wording "statistically similar". The test is to see if there is a difference at $\alpha = 0.5$, not if there is similarity at $\alpha = 0.95$. I would suggest stating that the number of NREM and REM bouts was "not significantly different".

We have changed the text accordingly.

"The proportion of NREM and REM sleep bouts were not statistically different for both recording sessions ($p_{NREM} S1/S2 = 0.17$, $p_{REM} S1/S2 = 0.65$; Fig. 1C)."

3- Fig 2. In the text there is no mention of the intense BOLD section in the most caudal coronal section (5th row third column). In the accompanying figure S5, this is the only coronal section where the nuclei are not delineated. It looks to me that this activity is in the cerebellovestibular process (pcv) and perhaps adjacent regions in the ventral part medial cerebellar nucleus (CbM), the medial part of the lateral cerebellar nucleus (CbL), and perhaps the dorsal part of the superior vestibular nucleus. These areas (pcv, medial CbL and ventral CbM) receive visual optic flow inputs from collaterals of mossy fibres originating in the nucleus of the basal optic root (Wylie et al. JCN, 1997). This lends support to the author's suggestion that the pigeons are dreaming about flying.

We thank the reviewer for the helpful comment. We have delineated the ROIs and added labels.

We have also changed the Results and Discussion sections as follows:

Results: "We also found activation clusters in the cerebellovestibular process (PCV), the ventral part of the medial cerebellar nucleus (CbM), and the medial part of the lateral cerebellar nucleus (CbL)²⁸"

Discussion: "The cerebellar nuclei, including PCV, medial CbL, and ventral CbM, were activated during REM sleep. They receive visual optic flow inputs from the collaterals of mossy fibers originating in the nucleus of the basal optic root²⁸."

4- page 10 line 5. It is stated here that the nucleus rotundas is activated by "optic flow". This is incorrect. The confusion may be that many neurons are responsive to looming objects on collision course, but Wang et al. emphasized that these were looming objects on a stationary background, but not looming surfaces.

Our reviewer is correct and we changed the sentence accordingly:

Discussion: "REM associated BOLD signals also covered those parts of the thalamic n. rotundus that are activated by looming objects on a stationary background⁴⁵. These rotundal signals are then processed in the pallial visual tectofugal system and its associative visual areas⁴⁶, which were also active during REM sleep. Thus, it is tempting to speculate that our pigeons might have dreamed about diverse scenes of flying."

5- page 10 lines 23-30. I find these statements to be rather speculative, and not supported by scientific findings. Can the authors reference any studies that suggest that the increase in blood flow does aid in waste removal?

We agree that this statement is speculative and we are not aware of any research showing that increased blood flow does or does not increase waste removal. Nonetheless, it is in keeping with the growing view that

hydrodynamic processes mediate waste removal from the brain (Rasmussen et al. *Physiol Rev*, 2022). Indeed, as the brain and CSF are enclosed by the rigid cranium, and changes in fluid volume in one region must be countered by a change in fluid in another, it seems reasonable to propose that the influx of blood into the brain (Bergel et al. *Nat Comm*, 2018), coupled with increased vascular diameter (Turner et al. *eLife*, 2020) and compression of the perivascular space through which CSF normally enters the brain (Bojarskaite et al. *bioRxiv*, 2022 - page 10, line 20), might squeeze CSF out from the perivascular space and through the extracellular space, thereby removing waste. Consequently, we think that it is reasonable and important to include this as a hypothesis. We strongly believe that our findings and the hypothesis proposed herein will inspire new research aimed understanding the potentially clinically relevant role that REM sleep plays in waste removal.

To more clearly alert the reader to the fact that we are proposing a novel mechanism by which REM sleep could enhance waste clearance, despite reducing CSF flow outside the brain, we modified the sentence as follows:

“However, at the same time, we propose that the increase in brain blood volume might squeeze the perivascular and extracellular spaces, thereby increasing flow through the brain tissue.”

6- page 13 line 7. The reference to Wylie (2013) is incorrectly listed as volume 0, but is actually volume 7. They refer to this study in regards to the activation of folium VI being concerned with optic flow. A more appropriate reference might be Wylie et al. (2018, *Frontiers in Neuroscience* 12:223) where the authors directly implicate the oculomotor cerebellum in flight through visually cluttered environments.

The reference was edited as suggested. In addition, we updated text to:

“When organisms move forward, they experience optic flow as a visual expansion in the direction of self-motion⁴¹, while local motion signals code for obstacles that should be avoided in flight through visually cluttered environments⁴².”

7- page 13 line 2; What do the authors mean by the birds were "ready" to be restrained? Could they offer a more descriptive operational definition? Was it simply that the birds readily fell asleep? Did some pigeons fail to habituate.

We adjusted the sentence as follows:

“After 18 days of habituation, all birds engaged in NREM and REM sleep in the training setup and proceeded to the next phase of obtaining the fMRI recordings.”

8- page 16, lines 10-15. How long to the birds remain awake after the presentation of the auditory stimulus?

To better understand the length of time the birds were awake following sound stimulation, we calculated the average power in the delta band (0.5-4Hz) for each second-long bin from the sound onset until 20 seconds after the sound offset. The values obtained for each second bin were then statistically tested against the average delta power values from the 10 s before stimulation. We modelled the data using linear mixed effect models, with the average delta power before and during each second-long bin after the stimulation onset as the response variable, the bin and trial number as a predictors, and the bird identity as a random effect. The delta power was significantly reduced for the 10 s following the onset of sound stimulation.

Second after stim onset	Contrast	Null value	Estimate	Std error	Statistic	Adj p value	Significance
1	T0 - T1	0	1.48E-10	4.17E-11	3.545	0.006	**
2	T0 - T2	0	2.43E-10	4.17E-11	5.839	8.46E-08	***
3	T0 - T3	0	2.76E-10	4.17E-11	6.626	1.98E-10	***

4	T0 - T4	0	2.71E-10	4.17E-11	6.505	7.50E-10	***
5	T0 - T5	0	2.58E-10	4.17E-11	6.190	4.59E-08	***
6	T0 - T6	0	2.23E-10	4.17E-11	5.341	1.33E-06	***
7	T0 - T7	0	2.06E-10	4.17E-11	4.936	1.52E-05	***
8	T0 - T8	0	2.17E-10	4.17E-11	5.218	2.29E-06	***
9	T0 - T9	0	1.81E-10	4.17E-11	4.332	0.000	***
10	T0 - T10	0	1.46E-10	4.17E-11	3.495	0.008	**
11	T0 - T11	0	1.08E-10	4.17E-11	2.600	0.112	
12	T0 - T12	0	8.44E-11	4.17E-11	2.025	0.375	
13	T0 - T13	0	1.14E-10	4.17E-11	2.737	0.079	.
14	T0 - T14	0	1.35E-10	4.17E-11	3.249	0.018	*
15	T0 - T15	0	1.45E-10	4.17E-11	3.479	0.008	**
16	T0 - T16	0	1.16E-10	4.17E-11	2.787	0.069	.
17	T0 - T17	0	6.83E-11	4.17E-11	1.639	0.668	
18	T0 - T18	0	3.94E-11	4.17E-11	0.945	0.995	
19	T0 - T19	0	1.14E-11	4.17E-11	0.273	1	
20	T0 - T20	0	1.83E-12	4.17E-11	0.044	1	

9- Figure captions for S4 and S6. I am confused. It is stated that green is day 1, and yellow is day 2. Is this correct? Or is it that day 1 in BLUE and day 2 is yellow and green represents the overlap?

We apologize for the confusion. We used blue and copper colors to represent the results from each session. We have changed the caption as follows:

“The areas with greater activation during REM sleep when compared to NREM sleep for session 1 (blue, n = 14) and session 2 (copper, n = 14). The activation patterns overlap (green) across large areas, indicating that the results are reproducible.”

Reviewer #2 (Remarks to the Author):

In this manuscript, the authors report two original findings both of which have significant relevance to the field of sleep. First, these kinds of explorations of bird sleep are fundamental to understanding human sleep. As the authors outline, birds, like mammals, express both REM and NREM sleep and these sub states share many electrophysiological and behavioral features. Despite the similarities, the states are not identical and sleep in birds evolved independently from mammalian sleep. This independent evolution suggests that sleep, and this two-state expression of it, must serve some fundamental function for the nervous system; if it weren't essential, evolution would have dispensed with. Instead, it seems, nature made only subtle modifications. In light of this, exploring the similarities and differences between mammalian and avian sleep, is an ideal way to reveal fundamental functions of NREM and REM sleep. Essentially, this type of comparative work allows us to see which features of sleep nature kept along the journey of evolution and to explore why they are so important.

The paper focuses on the role of sleep in glymphatic clearance in an avian model. The recently discovered glymphatic system is a CSF circulation pathway responsible for metabolic waste clearance; in mammals CSF circulation has been shown to increase during sleep in mammals and this increase is associated with clearance of toxins including, but not limited to, excess glutamate, lactate, and amyloid-beta. Although the vast majority of glymphatic clearance in mammals is thought to occur during sleep, recent data also suggests

that the process may also be modulated by the circadian system. Ultimately, the details of how metabolic waste products are cleared from the brain and the specific role mammalian sleep plays in this process is not entirely clear. A detailed understanding of sleep's role in this process has far-reaching importance, in humans, for physical and mental illnesses. By taking a comparative view of glymphatic function in sleeping birds, the authors provide evidence supporting a central role of NREM sleep in glymphatic activation. They also show something that has not been shown in a mammalian model; REM sleep does not support glymphatic clearance. The fact that this question has never been addressed in a mammalian system represents a major gap in the literature.

We appreciate the reviewer's recognition of the basic and clinical importance of understanding the impact that REM sleep has on CSF flow through studying birds.

The resting state BOLD data is also significant. Here the authors demonstrate that spontaneous brain activity in birds is shaped in a state specific way that largely recapitulates mammalian sleep (including humans). These data, and the fact that resting state data during sleep can be captured in birds without the use of anesthesia, have implications for our understanding the evolutionary principles of consciousness. Although a small number of very recent studies have successfully used functional neuroimaging in behaving birds, resting state data in awake animals is exceedingly difficult and, except in humans, has been almost exclusively collected under anesthesia—which significantly limits understanding of the brain's functional activation and architecture. Using the methodology described here, the authors can further explore the dynamic structure of the avian brain and its reconfiguration as a function of distinct states of consciousness (i.e. NREM unconsciousness vs REM consciousness) in a comparative way. In my view, the BOLD data shown here represent another step forward in the path towards dissolving our belief in mammalian exceptionalism.

Here again, we thank the reviewer for their appreciation of our work, as well as the minor comments that follow.

Although my view of the paper is overwhelmingly positive, below are minor questions related to methodology and interpretation.

1) As the authors describe, sleep structure and regulation in birds is quite similar to mammalian sleep, although REM and NREM states, as well as intermediate sleep (IS; dozy) bouts, are shorter and state transitions more frequent. My understanding of sleep in captive birds, informed in part by the careful work done by the last author (NR), is that when birds are not actively engaged in bird stuff (technical term encompassing the entire behavioral repertoire of the captive bird: eating/drinking, hopping/flapping, singing/calling, scratching/beak scraping) they are sleeping. One way to keep them from sleeping is to introduce unfamiliar sensory stimuli—the unusual musical selections used here would seem adequate. Given that the authors seem to have very clear data on eye closure, effectively tracking any unihemispheric sleep, why are the analyses largely restricted to REM vs NREM comparisons? They do compare to 'baseline' but they do not really make substantive comparisons with wake? If they plan to publish a distinct manuscript focused on glymphatic clearance as a function of behavioral state (sleep v wake) that should be mentioned. However, instead it seems that they are unable to clearly delineate NREM from wake. If so, what was the cause of this? Were bouts and transitions impacted by the inversion of the sleep period from night to day (i.e. was behavior, temporal distribution and bout duration, number of unihemispheric event unlike typical sleep)? Or are wake bouts generally too short during the sleep period to make meaningful comparisons with wake possible?

The reviewer makes several important points.

We initially chose not to examine unilateral eye closure and potential unihemispheric sleep for several reasons. First, although pigeons have been shown to sleep with one eye open for extended periods under certain conditions, the corresponding EEG asymmetry recorded from the visual hyperpallium at the dorsal surface of the brain is very small in pigeons (Rattenborg et al. Brain Behav Evol, 2001) when compared to other

birds, such as mallards (Rattenborg et al. *Nature*, 1999) and frigatebirds (Rattenborg et al. *Nat Comm*, 2016). Instead, the hemisphere connected to the open eye shows NREM sleep-related EEG slow-wave activity that is only slightly lower than that occurring in the hemisphere connected to the closed eye. For this reason, in subsequent studies of pigeons, we usually consider NREM sleep to be bihemispheric (e.g., Martinez-Gonzalez et al. *J Sleep Res*, 2008; Ungurean et al. *Curr Biol*, 2021). Consequently, we thought that a significant proportion of unilateral eye opening in the fMRI pigeons would be associated with slow-waves in the contralateral hyperpallium and thus, the corresponding brain activation pattern would not be representative of wakefulness.

For similar reasons we also did not delineate wakefulness from NREM sleep from behavior when the birds had both eyes open. In previous studies, it has been shown that many birds, including pigeons (Tobler and Borbély *J Comp Physiol, A*, 1988), can engage in NREM sleep with both eyes open, as defined by EEG activity in the visual hyperpallium. However, importantly, wakefulness never occurs when the birds are resting with their eyes closed (Rattenborg et al. *Principles and Practices of Sleep Medicine 7e*, 2022). Consequently, we initially took a conservative approach and focused on the two sleep states that have clear behavioral correlates: unequivocal periods of NREM sleep with both eyes closed and REM sleep, which also occurs with bilateral eye closure, but with the addition of rapid eye movements, rapid iris movements, and/or rapid bill movements.

Unfortunately, our lab is not equipped with an fMRI compatible EEG recording system. Consequently, to measure the impact that wakefulness has on CSF flow, we first established that a novel sound stimulus can be used to induce waking EEG activity. We then removed the EEG electrodes, and recorded the fMRI CSF flow response to the sound that previously elicited waking EEG activity. Using this approach, we were able to show that CSF is lower during sound-induced wakefulness than NREM sleep. Once we obtain funding for an expensive fMRI compatible EEG recording system, we intend to fully examine how brain activation and CSF flow changes during spontaneous wakefulness.

Nonetheless, in response to the Reviewer #2, #3 and #4's comments, we analyzed the fMRI signals occurring during periods with only one eye open or both eyes open. As outlined in the new section and figure (Figure 5), we found that when compared to NREM sleep with both eyes closed, birds with both eyes open showed significant activation primarily of the entopallium and nearby areas, all of which are involved in processing visual information received via the tectofugal pathway. Importantly, this activation was also present when the birds had only one eye open, with activation of the entopallium contralateral to the open eye being significantly greater than that contralateral to the closed eye. Consequently, it appears that the entopallium is awake when the corresponding eye is open. This is consistent with the finding that pigeons direct the open eye toward potential threats (Rattenborg et al. *Brain Behav Evol*, 2001).

Having discovered that eye opening is associated with activation in visual regions, we next examined whether having one or both eyes open had an influence on the CSF inflow signal. Importantly, we found that during all eye open states, the CSF signals in the IV ventricle and its cerebellar recess were significantly reduced when compared to that occurring during NREM sleep with both eyes closed. Consequently, as with the widespread activation of the brain occurring during REM sleep, activation of the entopallium associated with eye opening is also linked to reduced CSF flow. Interestingly, the reduction in CSF flow occurring during eye open states was not as pronounced as that occurring during REM sleep. This is likely due to the fact that the brain activation associated with eye opening was not as wide-spread as that occurring during REM sleep. Consequently, we expect that the CSF inflow signal would be reduced further when the pigeons are fully awake and interacting with their environment.

Obviously, we are quite pleased with these findings, and thank the reviewers for pushing us to examine our data in more detail! Indeed, we think that this analysis is far more informative than that focused on the sound stimuli, which was performed on a relatively small number of birds. Nonetheless, as it provides another window into wakefulness, we think the sound stimulation experiment should be retained in the manuscript.

Figure 5 - Association between eye open states and BOLD signal in the telencephalon (gray matter) and ventricular CSF inflow. GLM analysis was used to demonstrate the activated networks during eye opening by contrasting (A) both eyes open > NREM ($n = 9$), (B) left eye open > NREM ($n = 11$), and (C) the right eye open > NREM ($n = 12$) (group analysis using a FLAME1, $Z = 2.3$, and $p < 0.05$ FWE corrected to cluster level, group analysis). The functional maps were superimposed on the high-resolution anatomical data for the horizontal (left), axial (middle), and sagittal (right) axes of an ex vivo pigeon brain (in grayscale). The red to yellow scale shows areas with increased BOLD signal during eye open conditions, whereas the blue to green scale shows areas with increased fMRI signal during NREM sleep. Brain regions with increased signal were largely restricted to the entopallium or surrounding visual areas contralateral to the open eye(s), whereas areas with increased BOLD signal were largely restricted to the ventricular system. (D) To quantify the fMRI signal in these areas, we restricted the analysis to the entopallium (E, red area) and the IV ventricle and its cerebellar recess (IV and CR, blue areas). Estimated parameters from GLM analyzes for various conditions, including NREM, REM, left, right, and both eyes open, were extracted from voxels in E, IV and CR. (E) As with REM sleep, the BOLD signal in the entopallium was bilaterally elevated when both eyes were open; when only one eye was open, the contralateral entopallium showed significantly stronger BOLD signal than the ipsilateral entopallium (two-tailed paired t -test, $t_{Left}(20) = 2.03$, $p = 0.039$; $t_{Right}(22) = 1.93$, $p = 0.046$, corrected for multiple comparison). (F) During all eye open states and REM sleep, the fMRI signal in IV was decreased when compared to NREM sleep (two-tailed two-sample t -test, $t_{REM}(28) = 5.4$, $t_{Left}(24) = 3.6$, $t_{Right}(25) = 4.8$, $t_{Both}(22) = 3.1$, $p < 0.001$, corrected for multiple comparison). However, the estimated parameters were significantly lower during REM sleep when compared to the eye open states (two-tailed paired t -test, $t_{Left}(24) = -2.7$, $t_{Right}(25) = -2.5$, $t_{Both}(22) = -3.2$, $p < 0.01$, corrected for multiple comparisons). (G) similar patterns were observed in the CR.

Supplementary Table 1. List of subjects involved in the different analysis.

Animal Identity	REM and NREM ($n = 15$)	Left eye open ($n = 11$)	Right eye open ($n = 12$)	Both eyes open ($n = 9$)	Auditory stimulation ($n = 4$)	EEG and auditory stimulation out of scanner ($n = 6$)
29	✓	✓	✓	✓	×	×
33	✓	✓	✓	✓	✓	×
34	✓	✓	✓	✓	×	✓
35	✓	×	✓	✓	×	×
36	✓	✓	✓	✓	✓	×
37	✓	✓	✓	✓	×	×

38	✓	✓	✓	✓	✓	x
52	✓	✓	x	x	x	x
69	✓	✓	✓	x	✓	x
78	✓	✓	✓	✓	x	✓
84	✓	✓	✓	✓	x	✓
712	✓	x	✓	x	x	✓
721	✓	x	x	x	x	✓
735	✓	✓	✓	x	x	x
736	✓	x	x	x	x	✓

2) Given the brief duration of REM episodes and the frequent state transitions in birds, why was a TR of 4 seconds chosen to record whole-brain BOLD responses? Was there a reason for not choosing shorter (i.e 2 sec TR) sequences which are often used in task-related functional imaging paradigms in humans? Does this long TR have implications for the regressors associated with physiological noise (HR, respiration) in REM and NREM and their removal in the GLM?

Functional MRI in birds is challenging because of the anatomy of their skull, which is covered with large air cavities. These cavities lead to severe susceptibility artifacts. This limits the use of the gradient-echo EPI sequence (the most commonly used sequence for fMRI in humans, primates, and rodents). A spoiled gradient-echo fMRI sequence has already been used to image the auditory network in songbirds, but due to susceptibility artifacts, only 50% of the whole brain could be recorded (Van Meir et al. NeuroImage, 2005; Poirier and Van der Linden In vivo NMR Imaging: Methods and Protocols, 2011). To overcome this limitation (TR of 8 sec), our laboratory developed the single-shot RARE sequence (Behroozi et al. Nat Comm, 2020) with a TR of 4 s to further increase the temporal resolution and sensitivity of the original RARE protocol. But there is a limit to the temporal resolution of whole brain fMRI using RARE at 7T due to the heat within the animal body which the many refocussing RF pulses produce. Because the single-shot RARE sequence uses 42 refocusing 180° pulses per slice (11 slices) every 4 s. For this reason, we avoided shorter TR to further reduce stress by reducing gradient noise and the heat induced by the radiofrequency pulses.

We are aware of the effects of physiological noise such as respiration during longer TR. To account for artifacts associated with respiration, voxel-wise regressors for physiological noise based on respiratory signals were created using the PNM tool in FSL (Brooks et al. NeuroImage, 2008) by calculating respiratory phases relative to each volume and slice in the rs-fMRI signals.

3) The authors say that they “characterized the species-specific hemodynamic response function”. Do they mean they estimated activation parameters-lag time from state of interest, duration of bold signal, time to peak signal, max signal strength, from these animals? Or from existing literature? Is the change in maximum BOLD signal strength similar across brain regions and across behavioral state? If not, might this impact their ability to detect BOLD differences, for example, in the hippocampus in NREM vs REM? It is certainly conceivable that birds process memory differently than mammals, but could there be another explanation for the failure to see a mammalian-like hippocampal activation in REM.

Knowledge of the hemodynamic response function (HRF) is required to perform GLM analysis. For pigeon fMRI, we recently determined the HRF of pigeons (Behroozi et al. Nat Comm, 2020). Similar to human research, we used this HRF function to find the activated clusters. We are also aware that the HRF response can vary between brain regions. We therefore included the temporal derivative as an explanatory variable to account for the time delay between the regions.

To further rule out that activity in the hippocampal formation was not visible due to errors in data analysis, we did a ROI analysis to closely examine the hippocampal formation response during REM sleep. The results

showed that there is no BOLD signal increase following the REM sleep onset. Therefore, we can conclude that the function of the hippocampus during REM sleep in pigeons is different from that in mammals. The figure has been added to the supplement. We also elaborated on this point a bit more in the discussion.

“Regarding memory processing, the absence of a mammal-like activation of the hippocampal formation might be linked to the apparent absence of a hippocampal theta rhythm during avian REM sleep^{47,48}. As this rhythm is involved in processing memories during mammalian REM sleep⁴⁹, collectively, these findings suggest that birds might process hippocampal information differently from mammals during REM sleep⁴⁷.”

Supplementary Figure 5. The hippocampal formation signal during REM sleep.

The left panel shows the position of the pigeon hippocampal formation and the selected gray matter, previously shown to have increased BOLD signal during REM sleep, in three different views, axial, horizontal, and sagittal. The BOLD signal was extracted from voxels in the grey matter (red mask) and hippocampal formation (green mask). When compared to NREM sleep, the BOLD signal increased in the grey matter, but not the hippocampal formation during REM sleep. Error bars reflect the standard error of the mean across subjects.

4) What was the resolution of sleep state scoring? I assume it was continuous (not epoch based?)

Sleep was scored using 1-second epochs. Unlike other studies, we scored only pure-state while mixed epochs were left unscored.

To clarify this point, we have now added this information in the Methods:

“In all cases, a given state was attributed to an epoch, only when it occupied 100% of its duration.”

5) Noradrenergic cell groups in the pontine brainstem are conserved across fish, amphibians, reptiles, birds, and mammals, and play universal roles in modulating behavioral state, attention and stress. Although far less is known about the levels of brain norepinephrine (NE) during avian REM sleep, the conservation of the ascending arousal system would suggest that forebrain NE is similarly modulated in birds and mammals during NREM and REM sleep such that REM sleep is a state likely free of NE. Data in mammals suggests that NE plays a central role in glymphatic clearance. During mammalian sleep, the story goes, the decrease in NE levels causes the expansion of the extracellular space, decreasing resistance and increasing the rate of glymphatic clearance. Can the authors comment on how this story fits, or doesn't fit, with their data? Is there insufficient information on NE regulation in birds to conclude that it is a state void of NE? Differences

in glymphatic clearance during NREM vs REM have not yet been demonstrated in mammals, and this seems remarkable to me given the imputed importance of NE in the process.

We were quite pleased to read this comment. Indeed, in an earlier draft of the manuscript, we made this exact point. Certainly, if low NE was the main mediator of CSF flow through the brain during NREM sleep, then the even lower levels known to occur during REM sleep (at least in mammals) should increase flow even further. However, as the reviewer suggests, we are not aware of any studies that measured sleep-state dependent changes in NE or other neurotransmitters in birds. Consequently, we decided to leave this topic out of the discussion.

Nonetheless, as this point has relevance to our hypothesis on how REM sleep could influence CSF flow through the brain, we added the following italicized sentence:

“However, at the same time, we propose that the increase in brain blood volume might squeeze the perivascular and extracellular spaces, thereby increasing flow through the brain tissue. The high levels of acetylcholine (Ach) and low levels of norepinephrine (NE) occurring during REM sleep in mammals^{52,53} might facilitate this process, as Ach causes dilation of the cerebral arteries and a corresponding influx of blood⁵⁴, and low levels of NE increase extracellular space and the movement of CSF through the brain²; however, it is unknown whether Ach and NE levels change in the same manner in birds. Regardless, according to this model, most of the flow should accompany the surge of blood at the onset of REM sleep.”

6) As you will have guessed from my summary of the manuscript above, I find the BOLD data fascinating. The authors compellingly demonstrate that REM sleep is associated with activation in sensory motor and limbic regions in a mammalian-like way. When reflecting on these data in the Discussion they say, avoiding collisions with other objects.”

I assume these are captive birds? Existing data certainly supports the idea that birds are capable of higher-level consciousness-- as does common sense. Anyone who has ever lived near a red-winged black bird knows they exhibit a level of consciousness indistinguishable from that of the human bully; a show of weakness is grist for the mill. A natural extension of this higher ordered thinking is that birds have the ability to recapitulate, during sleep, previously lived experience. However, it's still hard to imagine that a bird might spontaneously imagine the experience of flight, including obstacle avoidance, and relive that experience during REM sleep.

We see the reviewer's point, but the pigeons were raised in groups housed in aviaries measuring 2m x 2m x 2m, and often had to avoid collisions with other birds when flying across the enclosure.

Reviewer #3 (Remarks to the Author):

This unusual and interesting paper examines CSF dynamics in the sleeping pigeon using fMRI. How sleep modulates brain fluid physiology is a topic of high interest, and very little prior work exists on REM sleep. This paper performed fMRI of sleeping pigeons and concludes that pigeon NREM sleep involves increased CSF flow (suggesting birds exhibit similar dynamics to humans) and that REM sleep involves decreased CSF and widespread increases in BOLD. The analysis methods are overall sensible, with some issues needing clarification. This work provides a novel report of sleep affecting CSF in birds, which are evolutionarily distant from humans, as well as one of the first reports of CSF imaging in REM sleep.

We thank the reviewer for their positive overview.

Comments:

- The sleep scoring was done manually using only the video recordings of the pigeons' pupil and head movements without any measurement of simultaneous EEG activity to confirm the sleep scores. It is true that EEG-fMRI is challenging, however, the authors had simultaneous EEG and pupil access outside the

scanner. It would be useful to know how accurate this approach is, which they could either validate in their own data, or specify from prior work. For example, it seems very plausible that birds are sometimes awake with their eyes closed, which means part of the NREM bouts would be wake bouts.

We appreciate the reviewer's point, but as noted above, we have the inverse problem; whereas wakefulness is not known to occur during eye closure, NREM sleep can occur with both eyes open or both eyes closed in pigeons (Tobler and Borbély J Comp Physiol, A, 1988), as well as many other birds (Rattenborg et al. Principles and Practices of Sleep Medicine 7e, 2022). Consequently, we initially focused on unequivocal NREM sleep with both eyes closed, and to assess changes in CSF flow during wakefulness, we presented a novel sound shown to induce wakefulness (i.e. EEG activation with both eyes open). As noted above, once we obtain funding for an expensive fMRI compatible EEG recording system, we intend to fully examine how brain activation and CSF flow changes during spontaneous wakefulness.

Nonetheless, we now include the EEG power spectra for behaviorally scored NREM and REM sleep, showing that as in previous studies, NREM behavior is associated with high power in the low frequencies, whereas REM sleep was associated with low power in these frequencies.

Supplementary Figure 4. Activity in left and right hyperpallial EEG corresponding to behaviorally defined sleep states and auditory stimulation (green).

Power spectral density was calculated for bouts scored as NREM (blue) and REM (red) sleep using the same behavioral criteria as for sleep scoring inside the fMRI scanner. The data is presented as mean \pm SEM.

- The Introduction states that they combine fMRI, EEG, and pupillometry, but EEG was not actually acquired during the fMRI measurements.

We see the reviewers point, and modified the text to avoid confusion:

“Here we combined fMRI, and pupillometry in awake and naturally sleeping pigeons to probe brain mechanisms underlying REM and NREM sleep.”

It seems that EEG was only acquired during an auditory stimulus, and the relevance of the finding that auditory stimuli elicit EEG responses in the pigeon is not clear. Could this be clarified? Why is a sound EEG experiment present in the supplement fig 8? It seems there is a concern that birds sometimes sleep with their eyes open and this would lead to mislabeling of the bouts.

Yes, as discussed elsewhere in our responses (see response to comment 1 of reviewer 2 for more details), it is known that pigeons and other birds can engage in NREM sleep with both eyes closed or both eyes open. Consequently, it is not possible to determine if a bird resting still with open eyes is awake or in NREM sleep.

For this reason, we only scored NREM sleep if both eyes were closed, and the birds were not exhibiting the behavioral signs of REM sleep; i.e. rapid eye movements, rapid iris movements (visible through the transparent eyelids), or rapid bill movements. Consequently, we used novel sound to induce clear awakenings, as shown in Fig. S3 and S9.

We have now included the power spectra for behaviorally scored NREM and REM sleep, showing the expected high power in the low frequencies for NREM sleep, and low power in these frequencies during REM sleep and wakefulness. In addition, as outline above for reviewer 2 and in a new section and figure (Fig. 5), we have now examined the fMRI signal when the birds had one or both eyes open. Importantly, we found that eye opening was associated with activation of the entopallium, suggesting that this visual region was awake. Furthermore, we found that this activation was associated with a reduction in the CSF inflow signal. Overall, this is consistent with the interpretation that brain activation occurring during REM sleep or wakefulness is associated with a reduction in CSF flow when compared to NREM sleep with both eyes closed.

However since EEG and auditory stimuli were not used in the main experiment, this control measure does not seem to help address the issue, and rather it seems to suggest that the wake bouts may be contaminated with sleep when acquired in the fMRI scanner?

As noted in the above response, we have now addressed this issue by analyzing the eye open states (see response to comment 1 of reviewer 2 for more details).

-Is it typical for pigeons to have as much REM as NREM or is this because the recordings were performed towards the end of the bird's perceived night?

Yes, it is common for birds to frequently cycle between NREM and REM sleep. Unrestrained pigeons can have over 700 REM sleep episodes per night (Tisdale et al. J Exp Biol, 2018). Also, as bouts of REM sleep are more frequent during the second half of the night in pigeons (Martinez-Gonzalez et al. J Sleep Res, 2008), we performed our recordings during this time to maximize the probability of recording large numbers of REM sleep episodes.

- How long were bouts? This information is important for interpreting the fMRI analysis, as the duration of bouts in the GLM will strongly affect its performance.

Histogram of REM sleep bout duration from all birds (n=15) and all sessions (n=29) combined.

As is typical for pigeons, and other birds (Rattenborg et al. Principles and Practices of Sleep Medicine 7e, 2022), bouts of REM sleep usually lasted < 10 s.

- In Fig 4b, the red and blue bars mark bouts of NREM and REM, but they have large gaps between them. What are the time gaps between bouts? Is it Wake data? If so, why would the pigeon wake up for prolonged periods between each bout? If not, why were those data not also modeled?

Only the unambiguous sleep bouts were scored and used in the analysis. Additionally, a minimum of two seconds at each state transition and all ambiguous states were omitted in the scoring to avoid including mixed states in the analysis. Ambiguous states include sleep with slow eye rolls under closed eyelids or one single eye movement without additional phasic events indicative of REM sleep.

“To further reduce variability and avoid including potential transitional states in the analysis, two seconds at state change and all ambiguous epochs were discarded.”

- Fig. 4 is very hard to read; I'd recommend putting line plots on a white background. The lines representing model fit of GM and model fit of CSF are not easy to see. Why is there a red dashed line in 4C? The legends are difficult to read.

We thank the reviewer for their recommendation and we have edited it to enhance clarity.

The red dashed line shows the negative peak of the cross-correlation between the gray matter BOLD and the CSF signal at a lag of +4 s. The information has been added to the figure caption.

- What do different colored circles represent in fig S9? It is unclear why there are only four of them.

Each circle represents a pigeon. Because the pigeons fell asleep immediately after head fixation (closing their eyes), we had to wake them up during the scan to compare wakefulness with sleep. For this purpose, we presented auditory stimuli. The auditory stimulation was done in a subset of four birds. Fig. S9 shows the comparison between the CSF signal during NREM and wakefulness in these four pigeons.

Unfortunately, we decided to do the auditory stimulation after completing the recordings of most of the birds. Consequently, this was only conducted on four birds. Although this auditory stimulation data is limited, we now include an analysis of eye open states (see above) on 11, 12, and 9 birds for the left, right and bilateral eye opening, respectively, to further examine the effect of wakefulness on the CSF inflow signal. We show that eye opening is associated with activation in the entopallium and a reduction in CSF inflow signal when compared to NREM sleep. Consequently, although the auditory stimulation data is limited, we feel that our new analysis of open eye states strongly links waking brain activity to reduced CSF flow.

- **“Thus, it is tempting to speculate that our pigeons might have dreamed about scenes of flying while avoiding collisions with other objects.” this is indeed extremely speculative.**

This statement is a speculation indeed, but it is based on the multiple evidence of activated regions. We found clusters of activation in the cerebellar nuclei cerebellovestibular process (PCV), the ventral part of the medial cerebellar nucleus (CbM), the medial part of the lateral cerebellar nucleus (CbL), and the dorsal part of the superior vestibular nucleus (SVN). These areas (PCV, medial CbL, and ventral CbM) receive visual optic flow inputs from collaterals of mossy fibers originating in the nucleus of the basal optic root (Wylie et al. J Comp Neurol, 1997). This supports our suggestion that pigeons might dream of flying (See Reviewer #1 comment 3 for more details).

- **It is not clear what the three colour bars in Supp Fig 3 represent. The legend states that Day 1 is green, and there is green in the image, but there is no green colour bar. The same issue is present in Supp Fig 5.**

We thank the reviewer for the remark. We have changed the legend as follows:

“Activation patterns of REM > NREM from session 1 (blue, n = 14) and session 2 (copper, n = 14). The activation patterns are overall very similar (green), indicating that the results are reproducible.”

Minor

- **supp fig 1: typo, ‘Fremwise’ should be ‘framewise’**

Changed.

Reviewer #4 (Remarks to the Author):

In this report, the authors describe different brain activations during REM and no-REM sleep states in pigeons, using fMRI, with widespread activations in the grey matter regions during REM and ‘activations’ in the ventricles during NREM . Technically, the experiment is impressive and the results are interesting in themselves. However, their interpretation is highly speculative, which leaves me with mixed feelings regarding the suitability of the manuscript to be published in this journal.

The goal of the study is too vague : the abstract and the introduction seem to suggest that the original goal was to test specific hypotheses related waste clearance but this aspect is not manipulated and towards the end of the introduction, the authors states was their goal was to ‘probe the brain mechanisms underlying REM and NREM sleeps’ . If the original goal was descriptive rather than hypothesis-driven, the authors should be transparent about it and the interpretation of fMRI activations in the ventricles in terms of CSF flow and waste clearance should be restricted to the discussion. Alternatively, if the authors’ intentions were to measure waste clearance/CSF flow by measuring fMRI signal in the ventricles, I would have expected some validation of the MRI signal, since it is unusual (fMRI signal in the ventricles are usually interpreted as artefacts and ignored).

The reviewer is correct that our initial plan was to compare brain activation during NREM and REM sleep in birds to that reported in mammals. However, as we were collecting the data, a paper was published in Science showing that CSF flow could be visualized using fMRI (Fultz et al. Science, 2019). We became even more interested in this topic when we found that nothing was known about CSF flow during REM sleep, likely due to the challenges of imaging REM sleep in mammals. Consequently, rather than automatically masking out the ventricles, we decided to also examine changes in the ventricular CSF signal between NREM and REM sleep. Importantly, instead of finding a constantly occurring CSF signal that might be interpreted as an artefact, we found a distinct CSF signal that depended on the bird’s brain state. We then interpreted this signal within the context of emerging views regarding the hydrodynamics occurring within the cranium. For these reasons, we think that the way the paper is laid out is reasonable. Indeed, we believe that it is very important to provide the reader with the background on what is and is not known about CSF flow upfront in the introduction.

Nonetheless, to be more transparent about the approach leading up to our findings, we changed the relevant text to:

“Here we combined fMRI, and pupillometry in awake and naturally sleeping pigeons to probe brain mechanisms underlying REM and NREM sleep. As we were collecting our data, Fultz and colleagues⁹ reported that ventricular CSF flow can be visualized using fMRI. Consequently, we also examined state-specific changes in the CSF signal.”

1- The authors need to do more to convince the reader that the birds are sleeping in the scanner, despite the noise induced by the fMRI sequence. What was the intensity of the fMRI sequence? An audio file of the auditory stimuli used to ‘wake up’ the birds with the background noise of the scanner would be helpful.

Prior to performing the fMRI experiments, the animals were habituated to head fixation and the magnet noise using the habituation protocol established in our laboratory for *in vivo* fMRI experiments (Behroozi et al. Nat Comm, 2020). In the scanner, habituated pigeons rapidly exhibited behaviors correlated with EEG-defined NREM sleep in previous studies and our EEG recordings of the pigeons during habituation; i.e. closure of both eyes, regular and slow breathing, and still eyes with dilated pupils (Ungurean et al. Curr Biol, 2021). These periods of behavioral NREM sleep were periodically followed by behaviors indicative of REM sleep (i.e. bilateral eye closure with rapid eye movements, rapid pupil constrictions, and/or rapid bill movements). Consequently, the pigeons’ behavior showed the typical repeated cycle between NREM and REM sleep. All these signs tell us that the animals fell asleep immediately after the head was fixed in the scanner. The sound pressure level of the single-shot RARE sequence noise (measured at 1 m from the magnet bore) was ~74 dB, while the maximum sound pressure level for auditory stimuli was about 90 dB (measured at 1 cm from the speaker). Given the intensity and novelty of the auditory stimulus, it was high enough to wake up the animals, as we showed using EEG signal in the mock scanner (Fig. S3 and S9) (see response to comment 1 of reviewer 3 for more details). This information was added to the manuscript.

In addition, as demonstrated in our new section and figure (Fig. 5) examining the relationship between eye open states and the fMRI signal, we clearly show that eye opening is associated with the activation of visual brain regions when compared to NREM sleep with the eyes closed. Moreover, we now show that, similar to REM sleep, this brain activation is associated with a reduction in the CSF inflow signal. Consequently, eye state and other behavioral signs of sleep/wake states clearly predicts changes in brain activation and CSF flow.

2- In human MRI experiments, it is common to see participants closing their eyes in the scanner if they do not have any task to do. However, only a minority of them report to have fallen asleep. Do the EEG data allow to exclude the possibility that the animals were in an ‘awake with eye closed’ state?

It is well established that eye closure in a bird at rest is a reliable indication of sleep, and no studies have found birds to be awake with their eyes closed (reviewed in, Rattenborg et al. Front Neurosci, 2019; Rattenborg et al. Principles and Practices of Sleep Medicine 7e, 2022). As demonstrated by several studies that examined the influence that predation risk has on eye closure and associated sleep-related brain activity in birds (e.g., Rattenborg et al. Nature, 1999), eye closure renders birds vulnerable to predation. Consequently, spending time awake with the eyes closed would be a maladaptive strategy, as the bird would forgo visual vigilance without gaining any of the benefits of sleep. Indeed, this likely explains why the inverse is often reported in birds; i.e., NREM sleep EEG activity recorded from the visual hyperpallium can occur with the eyes open in pigeons (Tobler and Borbély J Comp Physiol, A, 1988), and other birds (Rattenborg et al. Principles and Practices of Sleep Medicine 7e, 2022). We think that this strategy enables birds to obtain some of the benefits of sleep, while simultaneously allowing for the detection of threatening visual input. Indeed, our new analysis suggests that this wake-like responsiveness is mediated via activation on the visual entopallium.

Given that NREM sleep can occur with the eyes open when threatened, we conducted a careful habituation procedure to ensure that the birds felt comfortable enough to close their eyes. Furthermore, for our analysis of NREM sleep, we focused only on periods when the birds had both eyes closed. To confirm that this behavior corresponds to NREM sleep, we have now calculated the EEG power spectra for behaviorally scored NREM and REM sleep during the training sessions, and compared them to that occurring during the sound stimulation.

This analysis clearly shows that behaviorally scored NREM sleep was associated with high power in the low frequencies typical of NREM sleep, whereas REM sleep and the spectra during the sound stimulation showed low power in the low frequencies typical of REM sleep and wakefulness in pigeons (e.g., Tobler and Borbély J Comp Physiol, A, 1988; Martinez-Gonzalez et al. J Sleep Res, 2008), and other birds (e.g., Rattenborg et al. PLoS Biol, 2004).

3- I cannot make sense of the contrasts REM vs 'Baseline' and NREM vs 'baseline', since 'baseline' refers to a mixture of various states. The comparison of fMRI activations between REM vs awake states (as well as NREM vs. Awake) would be more useful (even if the auditory stimulation would influence the results).

We see the reviewer's point and have removed the comparisons to "baseline". In addition, as discussed above, we have added an analysis of eye open states which clearly shows a corresponding activation of visual regions in the core of the telencephalon when compared to NREM sleep. Importantly, this activation of visual brain regions was associated with a reduction in CSF flow when compared to NREM sleep. Although this analysis presumably does not show the full brain activation pattern that would occur in a fully awake pigeon interacting with its environment, it does demonstrate the impact that components of wakefulness have on CSF flow.

4- In humans, it is well established that activation of one brain region does not correspond to one single (perceptual or cognitive) process and therefore that the location of activations in the brain is not sufficient to infer the underlying experience of the participants. The interpretation of the results made by the authors is therefore highly speculative. Eye movements are also known to induce widespread brain activations in humans. I am therefore surprised that the authors did not discuss the possibility that (at least some of) their results observed during REM sleep simply reflect eye movements rather than 'perceptual' experience.

This is a good point, and we agree that comparing brain activation during phasic REM (with rapid eye movements) to that occurring during tonic REM sleep (without rapid eye movements) would be very interesting. However, we do not think that we can isolate the influence of eye movements, as they occur during the majority of the episodes of REM sleep, and when not present, other phasic events, such as rapid iris movements and/or bill movements occur. Indeed, birds seem to experience very little REM sleep that might be considered tonic (without phasic events), as described in mammals. In fact, during the short episodes of REM sleep that rarely exceed 10 s in birds, at best there are only a couple of seconds here and there that might be tonic. Given this interesting difference between REM sleep in mammals and birds, we do not think that this question can be addressed with fMRI.

References

- Behroozi, M., Helluy, X., Ströckens, F., Gao, M., Pusch, R., Tabrik, S., Tegenthoff, M., Otto, T., Axmacher, N., Kumsta, R., *et al.* (2020). Event-related functional MRI of awake behaving pigeons at 7T. *Nat Comm* 11, 4715.
- Bergel, A., Deffieux, T., Demené, C., Tanter, M., and Cohen, I. (2018). Local hippocampal fast gamma rhythms precede brain-wide hyperemic patterns during spontaneous rodent REM sleep. *Nat Comm* 9, 5364.
- Bojarskaite, L., Bjørnstad, D.M., Vallet, A., Binder, K.M.G., Cunen, C., Heuser, K., Kuchta, M., Mardal, K.-A., and Enger, R. (2022). Sleep cycle-dependent vascular dynamics enhance perivascular cerebrospinal fluid flow and solute transport. *bioRxiv*, 2022.2007.2014.500017.
- Brooks, J.C.W., Beckmann, C.F., Miller, K.L., Wise, R.G., Porro, C.A., Tracey, I., and Jenkinson, M. (2008). Physiological noise modelling for spinal functional magnetic resonance imaging studies. *NeuroImage* 39, 680-692.
- Fultz, N.E., Bonmassar, G., Setsompop, K., Stickgold, R.A., Rosen, B.R., Polimeni, J.R., and Lewis, L.D. (2019). Coupled electrophysiological, hemodynamic, and cerebrospinal fluid oscillations in human sleep. *Science* 366, 628-631.
- Louie, K., and Wilson, M.A. (2001). Temporally structured replay of awake hippocampal ensemble activity during rapid eye movement sleep. *Neuron* 29, 145-156.
- Malinowski, J.E., Scheel, D., and McCloskey, M. (2021). Do animals dream? *Conscious Cogn* 95, 103214.
- Manger, P.R., and Siegel, J.M. (2020). Do all mammals dream? *J Comp Neurol* 528, 3198-3204.
- Martinez-Gonzalez, D., Lesku, J.A., and Rattenborg, N.C. (2008). Increased EEG spectral power density during sleep following short-term sleep deprivation in pigeons (*Columba livia*): evidence for avian sleep homeostasis. *J Sleep Res* 17, 140-153.
- Medeiros, S.L.d.S., Paiva, M.M.M.d., Lopes, P.H., Blanco, W., Lima, F.D.d., Oliveira, J.B.C.d., Medeiros, I.G., Sequerra, E.B., de Souza, S., Leite, T.S., *et al.* (2021). Cyclic alternation of quiet and active sleep states in the octopus. *iScience* 24, 102223.
- Nieder, A., Wagener, L., and Rinnert, P. (2020). A neural correlate of sensory consciousness in a corvid bird. *Science* 369, 1626-1629.
- Pernet, C. (2014). Misconceptions in the use of the General Linear Model applied to functional MRI: a tutorial for junior neuro-imagers. *Front Neurosci* 8.
- Poirier, C., and Van der Linden, A.-M. (2011). Spin echo BOLD fMRI on songbirds. In *In vivo NMR Imaging: Methods and Protocols*, L. Schröder, and C. Faber, eds. (Totowa, NJ: Humana Press), pp. 569-576.
- Rasmussen, M.K., Mestre, H., and Nedergaard, M. (2022). Fluid transport in the brain. *Physiol Rev* 102, 1025-1151.
- Rattenborg, N.C., Amlaner, C.J., and Lima, S.L. (2001). Unilateral eye closure and interhemispheric EEG asymmetry during sleep in the pigeon (*Columba livia*). *Brain Behav Evol* 58, 323-332.
- Rattenborg, N.C., Lesku, J.A., and Libourel, P.-A. (2022). Sleep in nonmammalian vertebrates. In *Principles and Practices of Sleep Medicine 7e*, M.H. Kryger, ed. (Philadelphia: Elsevier), pp. 106–120.
- Rattenborg, N.C., Lima, S.L., and Amlaner, C.J. (1999). Half-awake to the risk of predation. *Nature* 397, 397-398.
- Rattenborg, N.C., Mandt, B.H., Obermeyer, W.H., Winsauer, P.J., Huber, R., Wikelski, M., and Benca, R.M. (2004). Migratory sleeplessness in the white-crowned sparrow (*Zonotrichia leucophrys gambelii*). *PLoS Biol* 2, e212.
- Rattenborg, N.C., Van Der Meij, J., Beckers, G.J.L., and Lesku, J.A. (2019). Local aspects of avian non-rem and rem sleep. *Front Neurosci* 13.
- Rattenborg, N.C., Voirin, B., Cruz, S.M., Tisdale, R., Dell'Omo, G., Lipp, H.P., Wikelski, M., and Vyssotski, A.L. (2016). Evidence that birds sleep in mid-flight. *Nat Comm* 7, 12468.

- Rößler, D.C., Kim, K., De Agrò, M., Jordan, A., Galizia, C.G., and Shamble, P.S. (2022). Regularly occurring bouts of retinal movements suggest an REM sleep-like state in jumping spiders. *PNAS* *119*, e2204754119.
- Tisdale, R.K., Lesku, J.A., Beckers, G.J.L., Vyssotski, A.L., and Rattenborg, N.C. (2018). The low-down on sleeping down low: pigeons shift to lighter forms of sleep when sleeping near the ground. *J Exp Biol* *221*, jeb182634.
- Tobler, I., and Borbély, A.A. (1988). Sleep and EEG spectra in the pigeon (*Columba livia*) under baseline conditions and after sleep deprivation. *J Comp Physiol, A* *163*, 729-738.
- Turner, K.L., Gheres, K.W., Proctor, E.A., and Drew, P.J. (2020). Neurovascular coupling and bilateral connectivity during NREM and REM sleep. *eLife* *9*, e62071.
- Ungurean, G., Martinez-Gonzalez, D., Massot, B., Libourel, P.-A., and Rattenborg, N.C. (2021). Pupillary behavior during wakefulness, non-REM sleep, and REM sleep in birds is opposite that of mammals. *Curr Biol* *31*, 5370-5376 e5374.
- Van Meir, V., Boumans, T., De Groof, G., Van Audekerke, J., Smolders, A., Scheunders, P., Sijbers, J., Verhoye, M., Balthazart, J., and Van der Linden, A. (2005). Spatiotemporal properties of the BOLD response in the songbirds' auditory circuit during a variety of listening tasks. *NeuroImage* *25*, 1242-1255.
- Wylie, D.R.W., Linkenhoker, B., and Lau, K.L. (1997). Projections of the nucleus of the basal optic root in pigeons (*Columba livia*) revealed with biotinylated dextran amine. *J Comp Neurol* *384*, 517-536.

REVIEWERS' COMMENTS

Reviewer #1 (Remarks to the Author):

All my concerns have been addressed by the authors. In my opinion this is an excellent study that makes a novel and important contribution to the literature

Reviewer #2 (Remarks to the Author):

The authors fully addressed the questions I raised in my initial review. I think the revised manuscript is improved and is certainly worthy of publication.

Reviewer #3 (Remarks to the Author):

The authors have addressed all my comments, congrats on the interesting paper!

Reviewer #4 (Remarks to the Author):

The authors have significantly improved the manuscript. In particular, the description of the results is clearer and easier to follow. However, I still think that the abstract, introduction and discussion sections are problematic.

Abstract: While each sentence is clear, they are not juxtaposed in a logical order. For instance, there is no logical connection between these two sentences: red: "But does waste clearance occur throughout sleep or is it specific to NREM sleep? Using fMRI of naturally sleeping pigeons, we show that REM sleep (...) is accompanied in birds with the activation of brain regions involved in processing visual information, including optic flow during flight.

Introduction: The current introduction clearly reflects a post-hoc aim of the study (justified by a key paper published when the authors were acquiring their data) rather than the original one. While I understand the authors' motivations to adjust the study goal, it results in an introduction difficult to follow for the reader. An easy way to fix the problem would be to start with the description of the original aim of the study (focus on brain activations during REM sleep in birds, which is interesting in itself), and in a second time, explained why they also took advantage of their data to investigate a second aspect (about waste clearance).

Discussion: As recognised by the authors in their answer to the reviewer's comment, they cannot distinguish whether the activation pattern observed during REM sleep is due to a perceptual (dream-like) experience or alternatively, from eye movements since both are confounded. The authors should incorporate this alternative interpretation in their manuscript to provide a more balanced discussion of their results.

Minor comment: In the color scale of fig 2 and 3, $p = 0.00$ is not plausible. Please replace by a meaningful value.

Reviewer #1 (Remarks to the Author):

All my concerns have been addressed by the authors. In my opinion this is an excellent study that makes a novel and important contribution to the literature.

Thanks!

Reviewer #2 (Remarks to the Author):

The authors fully addressed the questions I raised in my initial review. I think the revised manuscript is improved and is certainly worthy of publication.

Thanks!

Reviewer #3 (Remarks to the Author):

The authors have addressed all my comments, congrats on the interesting paper! *Thanks!*

Reviewer #4 (Remarks to the Author):

The authors have significantly improved the manuscript. In particular, the description of the results is clearer and easier to follow. However, I still think that the abstract, introduction and discussion sections are problematic.

Abstract: While each sentence is clear, they are not juxtaposed in a logical order. For instance, there is no logical connection between these two sentences: red: "But does waste clearance occur throughout sleep or is it specific to NREM sleep? Using fMRI of naturally sleeping pigeons, we show that REM sleep (...) is accompanied in birds with the activation of brain regions involved in processing visual information, including optic flow during flight.

Introduction: The current introduction clearly reflects a post-hoc aim of the study (justified by a key paper published when the authors were acquiring their data) rather than the original one. While I understand the authors' motivations to adjust the study goal, it results in an introduction difficult to follow for the reader. An easy way to fix the problem would be to start with the description of the original aim of the study (focus on brain activations during REM sleep in birds, which is interesting in itself), and in a second time, explained why they also took advantage of their data to investigate a second aspect (about waste clearance).

The problems with the abstract have been addressed by the editorial team. As the abstract mirrors the overall structure of our Introduction, we do not think that it should be changed as requested by reviewer 4. Nonetheless, we made some changes to make it clearer that we decided to focus on CSF flow mid-way through the study.

Discussion: As recognised by the authors in their answer to the reviewer's comment, they cannot distinguish whether the activation pattern observed during REM sleep is due to a perceptual (dream-like) experience or alternatively, from eye movements since both are confounded. The authors should incorporate this alternative interpretation in their manuscript to provide a more balanced discussion of their results.

We have expanded the first paragraph of the Discussion to address the potential role eye-movements play in the observed activation patterns. We think this has strengthened the Discussion.

Minor comment: In the color scale of fig 2 and 3, $p = 0.00$ is not plausible. Please replace by a meaningful value.

Thanks for catching this error. It has been corrected.